# Unlearning during Training: Domain-Specific Gradient Ascent for Domain Generalization

**Di Zhao**[1], **Jingfeng Zhang**[1], **Hongsheng Hu**[2], **Philippe Fournier-Viger**[3],
**Gillian Dobbie**[1] **& Yun Sing Koh**[1]
[1]University of Auckland, Auckland, New Zealand
[2]Shanghai Jiao Tong University, Shanghai, China
[3]Shenzhen University, Shenzhen, China
`{di.zhao, jingfeng.zhang, g.dobbie, y.koh}@auckland.ac.nz`
`hongsheng.hu@sjtu.edu.cn, philfv@szu.edu.cn`

## Abstract

Deep neural networks often exhibit degraded performance under domain shifts due to reliance on domain-specific features. Existing domain generalization (DG) methods attempt to mitigate this during training but lack mechanisms to adaptively correct domain-specific reliance once it emerges. We propose Identify and Unlearn (IU), a model-agnostic module that continually mitigates such reliance post-epoch. We introduce an unlearning score to identify training samples that disproportionately increase model complexity while contributing little to generalization, and an Inter-Domain Variance (IDV) metric to reliably identify domain-specific channels. To suppress the adverse influence of identified samples, IU employs a Domain-Specific Gradient-Ascent (DSGA) procedure that selectively removes domain-specific features while preserving domain-invariant features. Extensive experiments across seven benchmarks and 16 DG baselines show that IU consistently improves out-of-distribution generalization, achieving average accuracy gains of up to 3.0%.

## 1 Introduction

The success of deep neural networks often relies on the assumption that training (source domain) and testing (target domain) data are drawn from the same distribution. However, this assumption is frequently violated in real-world scenarios due to domain shifts (Wang et al., 2022). Domain adaptation (DA) methods address this issue by adapting knowledge from a source to target domains with limited labeled target data (Pan & Yang, 2009). Unsupervised domain adaptation (UDA) further relaxes this requirement by leveraging only unlabeled data from the target domain (Xu et al., 2019). Despite their effectiveness, DA and UDA methods require access to target domain data during training, limiting their practicality (Yue et al., 2019). Domain generalization (DG) aims to overcome this constraint by training models on multiple labeled source domains to generalize to unseen target domains. Existing methods can be divided into three categories: data augmentation (Zhou et al., 2020), representation learning (Wang et al., 2022), and training strategies (Zhao et al., 2025).

A key limitation of existing DG approaches is their reliance on training-time objectives to prevent the learning of domain-specific features, without any mechanism to correct such reliance once it emerges. As a result, when models inadvertently capture domain-specific features during training, these methods lack the capability to remove them. Importantly, domain-specific reliance may arise dynamically at different stages of training (Piratla et al., 2020), suggesting the importance of an adaptive procedure that operates continually. This raises the central research problem of designing an adaptive mechanism that identifies training samples introducing domain-specific biases and then removes the associated features, while preserving domain-invariant features.

To address this problem, we propose Identify and Unlearn (IU), a model-agnostic module that continually mitigates reliance on domain-specific features as they emerge during training. IU adaptively intervenes after each epoch to detect and remove domain-specific features acquired by the model. The core idea builds on influence functions (Koh & Liang, 2017) and a principled view of general-

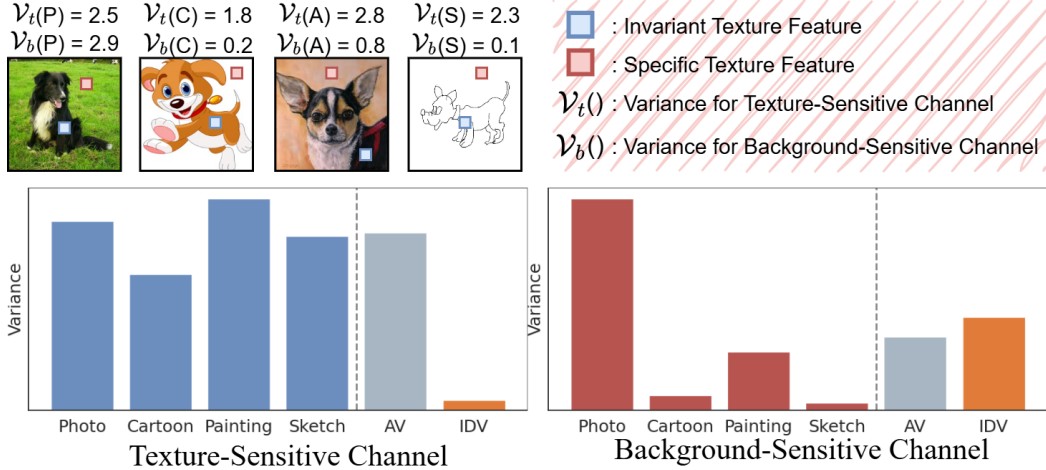

Figure 1: Illustration of Aggregated Variance (AV) versus Inter-Domain Variance (IDV). A texture-sensitive channel that responds to fur edges exhibits consistently high activation variance within each domain—natural fur in photos, bold outlines in cartoons, brush-stroke textures in paintings, and simplified contour strokes in sketches. AV, which pools variances across samples, incorrectly assigns this channel a high score and mislabels it as domain-specific, whereas IDV correctly identifies it as domain-invariant. A background-sensitive channel exhibits domain-dependent variability—large in photos, moderate in paintings, lower in cartoons, and minimal in sketches. AV underestimates this cross-domain variability, while IDV accurately flags it as domain-specific.

ization (Zhang et al., 2024): training samples that increase model complexity without contributing to generalization are likely to introduce domain-specific bias. IU identifies such samples and selectively suppresses their domain-specific features while preserving domain-invariant features. Specifically, IU introduces a novel unlearning mechanism that (i) identifies an unlearning set of training samples that disproportionately increases model complexity while contributing little to generalization, (ii) pinpoints the domain-specific channels, and (iii) removes the domain-specific features of the unlearning set by selectively reversing their gradients on the domain-specific channels.

To identify the unlearning set, we introduce the unlearning score, which highlights samples that disproportionately increase model complexity while contributing little to generalization. To identify domain-specific channels, we propose Inter-Domain Variance (IDV), a domain-aware metric that explicitly quantifies cross-domain variability. IDV first computes channel-wise activation variance within each domain, and then measures the variance of these variances across domains. Channels with high IDV values are therefore deemed domain-specific. This design makes IDV naturally robust to domain imbalance and less susceptible to conflating uniformly noisy channels with genuinely domain-specific ones. In contrast, existing approaches such as Aggregated Variance (AV) (Guo et al., 2023; Yu et al., 2024) suffer from two key limitations. First, AV is highly sensitive to domain imbalance, often yielding biased estimates. Second, AV primarily captures within-domain variance rather than cross-domain variability, which can lead to systematic misidentification. For example, a texture-sensitive channel that responds to fur edges exhibits consistently high variance across domains. Despite being domain-invariant, AV assigns this channel a high score and incorrectly identifies it as domain-specific (Figure 1). By directly capturing cross-domain variability, IDV offers a more principled and reliable criterion for domain-specific channel identification.

Our contributions are threefold. (1) We introduce a novel paradigm for DG that adaptively removes domain-specific features post-epoch, enhancing the robustness of existing DG methods. (2) We propose Inter-Domain Variance (IDV), a principled domain-aware metric that quantifies cross-domain variability, enabling reliable identification of domain-specific channels. (3) We develop Identify and Unlearn (IU), a model-agnostic module that adaptively identifies the unlearning set and removes their influence through Domain-Specific Gradient-Ascent. This selective unlearning mechanism mitigates domain-specific bias while preserving domain-invariant features. Extensive experiments across seven benchmarks and fifteen DG baselines demonstrate that IU consistently improves generalization to unseen domains, achieving average accuracy gains of up to 3.0%.

## 2 RELATED WORK

**Domain Shift** refers to performance degradation caused by discrepancies between source (training) and target (testing) domain distributions (Pan & Yang, 2009). To mitigate this issue, Domain Adaptation (DA) methods align distributions between source and target domains (Baktashmotlagh et al., 2013; Luo et al., 2020) or fine-tune models using annotated data from the target domain (Long et al., 2015). Despite significant progress, DA methods are practically limited by their dependency on annotated target domain data, which can be difficult to obtain (Yue et al., 2019). Unsupervised Domain Adaptation (UDA) addresses this limitation by transferring knowledge from labeled source domains to unlabeled target domains, eliminating the need for target domain annotations (Liu et al., 2021). However, UDA still necessitates data collection and adaptation for each target domain. This constraint motivates the development of approaches capable of generalizing effectively to unseen domains without relying on target domain data during training (Wang et al., 2022; Qiao et al., 2026).

**Domain Generalization** (DG) was initially introduced by (Blanchard et al., 2011) and subsequently formalized by (Muandet et al., 2013). Existing DG methods typically fall into three categories: data augmentation, representation learning, and training strategies. Data augmentation methods enhance dataset diversity through techniques such as domain randomization (Yue et al., 2019), adversarial augmentation (Shankar et al., 2018), and feature interpolation (Sun et al., 2022; Zhang et al., 2022). Representation learning methods seek to disentangle domain-invariant from domain-specific features via methods such as feature alignment (Jin et al., 2020), adversarial learning (Zhao et al., 2020), or invariant risk minimization (Ahuja et al., 2021a;b). Training strategy methods optimize model generalizability through different training strategies, including ensemble learning (Dubey et al., 2021), meta-learning (Balaji et al., 2018), self-supervised learning (Jeon et al., 2021), and curriculum learning (Zhao et al., 2024). DG has proven effective across various domains, including re-identification (Li et al., 2025; Wu et al., 2026). A key limitation of existing DG approaches is their reliance on training-time objectives to prevent the learning of domain-specific features, without any mechanism to correct such reliance once it emerges. To the best of our knowledge, IU is the first framework that explicitly removes domain-specific features after they have been learned.

**Machine Unlearning** (MU) was initially proposed to enable machine learning models to satisfy data deletion requests without requiring full retraining from scratch (Cao & Yang, 2015; Bourtoule et al., 2021). The concept of MU has subsequently extended across various machine learning domains, including image classification (Jia et al., 2023), image generation (Fan et al., 2024), graph neural networks (Wu et al., 2023), and large language models (Yao et al., 2024). Existing MU methodologies broadly encompass exact unlearning through retraining (Thudi et al., 2022), differential privacy-based unlearning (Sekhari et al., 2021), and approximate unlearning via fine-tuning Jia et al. (2023). Beyond data privacy, recent works have explored MU to enhance model performance in tasks such as continual learning (Wang et al., 2024) and domain adaptation (Basak & Yin, 2024). However, the potential of MU to improve generalization to unseen target domains remains largely unexplored. Crucially, DG presents a fundamentally different challenge from traditional MU applications: rather than forgetting specific data points for safety, the objective is to selectively unlearn domain-specific features that impede generalization. This key distinction motivates our work, which, to the best of our knowledge, is the first to explore machine unlearning as a tool for enhancing DG.

## 3 IDENTIFY AND UNLEARN FOR DOMAIN GENERALIZATION

We first describe the procedure for selecting the unlearning set (Section 3.1). We then introduce our approach for identifying domain-specific channels (Section 3.2), followed by the formulation of Domain-Specific Gradient Ascent (Section 3.3). Section 3.4 presents a theoretical analysis of the proposed framework. An overview of the IU module within a single post-epoch is illustrated in Figure 2, and the full algorithm is provided in Appendix D.

### 3.1 UNLEARNING SET SELECTION

Motivated by the principle that, among models with comparable training performance, simpler models tend to generalize better (Neyshabur et al., 2017; Zhang et al., 2024), we aim to identify and unlearn training samples that disproportionately increase model complexity while contributing minimally to generalization. To this end, we adopt the norm of the model parameters (Neyshabur et al.,

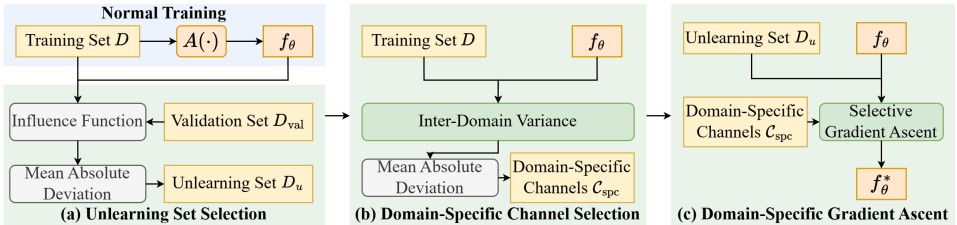

Figure 2: Overview of the IU module within a single post-epoch. (a) **Unlearning Set Selection**: Following standard training, the influence function is applied to the training set $D$ to estimate each sample's impact on model complexity and generalizability. Samples are then selected for unlearning via Median Absolute Deviation (MAD) thresholding, forming the unlearning set $D_u$. (b) **Domain-Specific Channel Selection**: Inter-Domain Variance (IDV) is computed across the training set to quantify variability in within-domain activation variances. Domain-specific channels $\mathcal{C}_{\text{spc}}$ are also identified using MAD thresholding. (c) **Domain-Specific Gradient Ascent**: Gradient ascent is selectively applied to domain-specific channels using samples from $D_u$, yielding the updated model $f_\theta^*$ with reduced domain-specific features and improved generalization capability.

2017) to quantify model complexity and leverage the influence function (Koh & Liang, 2017) to estimate the impact of each training sample on both model parameters and validation performance. Specifically, we define two metrics: a complexity score and a generalization score. The complexity score, $C_x$, defined in Equation 1, quantifies the impact of removing a single training sample $x$ on the model's complexity. This is measured by the $\ell_2$-norm of the resulting change in model parameters, serving as a proxy for model complexity. A higher complexity score indicates a greater influence of the sample on model complexity. Here, $\theta$ denotes the model parameters, $H_\theta^{-1}$ is the inverse Hessian matrix, and $\nabla_\theta L(x, \theta)$ represents the gradient of the loss function with respect to $\theta$. Since we are concerned only with the magnitude of parameter changes, not their direction, the $\ell_2$-norm is employed. This reduces the high-dimensional vector $-H_\theta^{-1}\nabla_\theta L(x, \theta)$ to a scalar, enabling direct comparison of complexity contributions across training samples.

$$C_x = \| -H_\theta^{-1}\nabla_\theta L(x, \theta) \|_2 \tag{1}$$

The generalization score, $G_x$, defined in Equation 2, quantifies the influence of a training sample $x$ on the model's performance over the validation set $D_{\text{val}}$, with higher values indicating a greater positive impact on generalization. Specifically, $G_x$ is a scalar value computed by aggregating the influence of $x$ across validation samples. Here, $-\nabla_\theta L(z, \theta)^\mathsf{T}$ represents the transpose of the gradient of the loss function with respect to the model parameters $\theta$, evaluated at a validation sample $z$.

$$G_x = \sum_{z \in D_{\text{val}}} -\nabla_\theta L(z, \theta)^\mathsf{T} H_\theta^{-1}\nabla_\theta L(x, \theta) \tag{2}$$

After computing the complexity and generalization scores, we define an overall unlearning score $U_x$ for each training sample $x$, as shown in Equation 3. This score guides the selection of samples for the unlearning set through a threshold-based criterion. A lower $U_x$ indicates that a sample contributes little to generalization while substantially increasing model complexity, making it a suitable candidate for unlearning. The parameter $\alpha$ mitigates **Score Equivalence Bias**: without it, the unlearning score may assign equal importance to samples with markedly different generalization and complexity scores, thereby reducing the effectiveness of the selection process. A more detailed sensitivity analysis is provided in Appendix F

$$U_x = \frac{(G_x)^\alpha}{C_x} \tag{3}$$

We then compute a threshold $\tau_U$, selecting as the unlearning set those samples with unlearning scores below $\tau_U$. To determine $\tau_U$, we adopt Median Absolute Deviation (MAD) thresholding (Equation 4). In this formulation, $U_{\tilde{x}}$ denotes the median of all unlearning scores, and $k$ controls the strictness of the threshold: higher values of $k$ result in fewer samples being selected for unlearning. Unlike alternative approaches such as mean absolute deviation, MAD does not assume any specific underlying distribution and is inherently more robust to outliers. Following standard practice in robust thresholding, we set $k = 2$ in all experiments.

$$\tau_U = U_{\tilde{x}} - k \times \text{median}(|U_{x_i} - U_{\tilde{x}}|, \quad x_i \in D_{\text{train}}) \tag{4}$$

## 3.2 DOMAIN-SPECIFIC CHANNEL SELECTION

After selecting the unlearning set, it is essential to selectively remove its impact on the model. A naive approach would be to unlearn all knowledge associated with the unlearning set; however, this is suboptimal, as it would also erase domain-invariant features crucial for generalization. To address this, we propose a targeted strategy that first identifies domain-specific channels, which are predominantly responsible for capturing domain-specific features. By focusing on these channels, our method selectively removes domain-specific features associated with the unlearning set, while preserving domain-invariant features that benefit generalization.

Several studies have investigated the relationship between channel activation variability and domain-specific features (Xu et al., 2022; Guo et al., 2023; Yu et al., 2024). However, prior methods typically measure channel-wise variability by computing the aggregated variance across activations pooled from all source domains, a process referred to as aggregated variance. These approaches implicitly assume that domain-specific channels exhibit high overall variability across the dataset, regardless of domain boundaries. This assumption conflates within-domain dispersion with between-domain differences and is particularly susceptible to biases under domain imbalance, where dominant domains disproportionately influence the aggregated statistics. As a result, channels may be erroneously classified as domain-invariant or domain-specific based solely on the magnitude of within-domain variation in the dominant domains.

In contrast to prior approaches, we propose a more principled definition of domain-specificity that more accurately reflects the structure of domain shifts. Specifically, we define a channel as domain-specific if its activation variance within domains differs substantially across source domains. To quantify this, we introduce a novel metric, Inter-Domain Variance (IDV), which measures the variance of a channel's within-domain variances across all source domains. Compared to aggregated variance, IDV offers two advantages: (1) It is domain-aware, treating each domain as an independent unit of analysis, and (2) It is domain-size agnostic, assigning equal importance to each domain regardless of its size. Formally, the IDV for a given channel $c$ is defined in Equation 5. Here, $N$ denotes the number of source domains, and $v_{(d)}^c$ represents the variance of channel $c$ within domain $d$. The term $x_{(d,i)}^c$ represents the activation of the $i$-th sample in channel $c$ from domain $d$, $\mu_{(d)}^c$ is the mean activation of channel $c$ within domain $d$, and $N_d$ is the number of samples in domain $d$.

$$\text{IDV}(c) = \text{Variance}\left(\left\{v_{(d)}^c\right\}_{d=1}^{N}\right), v_{(d)}^c = \frac{1}{N_d}\sum_{i=1}^{N_d}\left(x_{(d,i)}^c - \mu_{(d)}^c\right)^2 \tag{5}$$

By focusing on across-domain variation rather than within-domain variation, IDV offers a more precise characterization of domain-specific channels, mitigating biases introduced by imbalanced domain distributions. To identify domain-specific channels, we adopt MAD thresholding, consistent with the strategy used for unlearning set selection. Channels with IDV values exceeding this threshold are selected as domain-specific.

## 3.3 DOMAIN-SPECIFIC GRADIENT ASCENT

After identifying the unlearning set $D_u$ and the set of domain-specific channels $\mathcal{C}_{\text{spc}}$, IU selectively removes domain-specific features via Domain-Specific Gradient Ascent, as defined in Equation 6. Specifically, $\theta^c$ denotes the parameters associated with domain-specific channel $c$, and $\nabla_{\theta^c}L(x,\theta)$ represents the gradient of the loss with respect to $\theta^c$ for training sample $x$.

$$\theta^c = \theta^c + \nabla_{\theta^c}L(x,\theta), \quad x \in D_u, \quad c \in \mathcal{C}_{\text{spc}} \tag{6}$$

By selectively applying gradient ascent on domain-specific channels using data from the unlearning set, IU removes domain-specific features while preserving domain-invariant features.

## 3.4 THEORETICAL ANALYSIS

This section presents a theoretical analysis demonstrating that our approach reduces the model's predictive reliance on domain-specific features while preserving the domain-invariant features.

**Assumption 1** (Feature Decomposition). *Following standard assumptions in domain generalization and domain adaptation, we assume that the learned representation can be decomposed into domain-*

*invariant and domain-specific components:*

$$f(x) = [f_{\mathrm{inv}}(x), f_{\mathrm{spc}}(x)] \tag{7}$$

*where $f_{\mathrm{inv}}(x)$ denotes domain-invariant features and $f_{\mathrm{spc}}(x)$ denotes domain-specific features.*

**Definition 1** (Parameter Decomposition). *Let $\theta = (\theta_{\mathrm{inv}}, \theta_{\mathrm{spc}})$ be the corresponding parameters, where $\theta_{\mathrm{inv}}$ only influences $f_{\mathrm{inv}}(x)$ and $\theta_{\mathrm{spc}}$ only influences $f_{\mathrm{spc}}(x)$.*

**Definition 2** (Predictive Dependence). *Let $D_{\mathrm{inv}}(\theta) := I(y; f_{\mathrm{inv}}(x))$ denote the mutual information between the model output and domain-invariant features. Similarly, let $D_{\mathrm{spc}}(\theta) := I(y; f_{\mathrm{spc}}(x))$ denote the mutual information between the model output and domain-specific features.*

**Theorem 1** (DSGA Selectively Reduces Dependence on Domain-Specific Features). *Let $\theta^t$ be the model parameters at step $t$. Then, the update:*

$$\theta_{\mathrm{spc}}^{t+1} = \theta_{\mathrm{spc}}^t + \nabla_{\theta_{\mathrm{spc}}} \mathbb{E}_{x \in D_u}[L(x, \theta)] \tag{8}$$

*with $\theta_{\mathrm{inv}}^{t+1} = \theta_{\mathrm{inv}}^t$ yields:*

$$D_{\mathrm{spc}}(\theta^{t+1}) < D_{\mathrm{spc}}(\theta^t), D_{\mathrm{inv}}(\theta^{t+1}) \approx D_{\mathrm{inv}}(\theta^t) \tag{9}$$

*In other words, DSGA reduces the model's reliance on domain-specific features while preserving its dependence on domain-invariant features.*

*Proof.* Let the gradient ascent update be applied to domain-specific parameters:

$$\theta_{\mathrm{spc}}^{t+1} = \theta_{\mathrm{spc}}^t + \nabla_{\theta_{\mathrm{spc}}} \mathbb{E}_{x \in D_u}[L(x, \theta)] \tag{10}$$

Under the chain rule:

$$\nabla_{\theta_{\mathrm{spc}}} L(x, \theta) = \left( \frac{\partial f_{\mathrm{spc}}(x)}{\partial \theta_{\mathrm{spc}}} \right)^{\top} \cdot \nabla_{f_{\mathrm{spc}}} \ell(g([f_{\mathrm{inv}}(x), f_{\mathrm{spc}}(x)]), y), \tag{11}$$

the update affects only the influence of $f_{\mathrm{spc}}$ on the model output.

After this update, define the perturbed prediction:

$$\hat{y}^{t+1} = g([f_{\mathrm{inv}}(x), f_{\mathrm{spc}}(x; \theta_{\mathrm{spc}}^{t+1})]) \tag{12}$$

We now assess how this update affects the mutual information $D_{\mathrm{spc}} = I(y; f_{\mathrm{spc}}(x))$.

By properties of mutual information:

$$I(y; f_{\mathrm{spc}}(x)) = H(y) - H(y \mid f_{\mathrm{spc}}(x)) \tag{13}$$

where $H(y)$ is the entropy of labels and is fixed. So, reducing mutual information is equivalent to increasing the conditional entropy $H(y \mid f_{\mathrm{spc}}(x))$, i.e., making the prediction less certain given $f_{\mathrm{spc}}$. The gradient ascent update increases the expected loss on $D_u$, thereby decreasing the model's predictive confidence (i.e., increasing entropy) conditioned on $f_{\mathrm{spc}}$. Formally, this means:

$$H(y \mid f_{\mathrm{spc}}(x); \theta^{t+1}) > H(y \mid f_{\mathrm{spc}}(x); \theta^t) \tag{14}$$

implying:

$$I(y; f_{\mathrm{spc}}(x)) \downarrow \tag{15}$$

Since $\theta_{\mathrm{inv}}$ is not updated, the functional form and behavior of $f_{\mathrm{inv}}(x)$ and its contribution to the prediction remain unchanged, so:

$$\mathcal{D}_{\mathrm{inv}}(\theta^{t+1}) \approx \mathcal{D}_{\mathrm{inv}}(\theta^t) \tag{16}$$

Therefore, the DSGA update reduces $\mathcal{D}_{\mathrm{spc}}(\theta)$, i.e., the model's predictive dependence on domain-specific features, while preserving its predictive dependence on domain-invariant features. $\qquad \square$

# 4 EXPERIMENTS

## 4.1 EXPERIMENT SETTING

We evaluate our proposed module on the DomainBed benchmark (Gulrajani & Lopez-Paz, 2021) using five widely adopted datasets: PACS, OfficeHome, VLCS, Terra Incognita, and DomainNet. Additionally, we assess generalization performance on Digits-DG, following the protocol of (Zhou et al., 2020), and on NICO++, following the setup in (Zhang et al., 2023). To demonstrate both the effectiveness and model-agnostic nature of our unlearning module, we integrate it with a diverse set of state-of-the-art domain generalization baselines spanning multiple methodological categories. Consistent with previous studies (Gulrajani & Lopez-Paz, 2021; Zhou et al., 2022), we adopt a leave-one-domain-out evaluation protocol. Details on the baseline methods, evaluation metrics, validation set selection, and implementation settings are provided in Appendix C.

## 4.2 EXPERIMENTAL RESULTS

Table 1 reports the accuracy of the most up-to-date baselines across all benchmarks. Detailed results for additional baselines are available in Appendix K. The best-performing results are highlighted in bold. We evaluate two versions of our proposed module: the standard Identify and Unlearn (IU), denoted as the baseline with the subscript IU, and a variant incorporating Exponential Moving Average (EMA) smoothing to the unlearning score, denoted as the baseline with the subscript IUE. A detailed analysis of the effect of EMA smoothing is provided in Section 4.4.

We draw three key observations from Table 1. (1) IU enhances the generalization performance of all baselines, regardless of their underlying methodology, underscoring both its model-agnostic nature and the effectiveness of removing domain-specific features through post-epoch unlearning. (2) Applying EMA to smooth the unlearning score yields further gains, highlighting the importance of stabilizing the unlearning process. (3) Even for strong baselines, such as UDIM and VL2V, whose performance has plateaued on standard benchmarks, IU delivers modest yet consistent improvements, further demonstrating its ability to enhance the generalizability of existing DG methods.

## 4.3 ABLATION STUDY

Table 2 presents the results of an ablation study evaluating the individual contributions of Unlearning Set Selection (USS) and Domain-Specific Channel Selection (DSCS). The second row ($ERM_{USS}$) reports the performance when only USS is applied: gradient ascent is performed over the entire parameter set using the unlearning set, without distinguishing between domain-specific and domain-invariant features. As shown, indiscriminately reversing gradients for the unlearning set degrades generalization performance, as it inadvertently removes domain-invariant knowledge. The third row ($ERM_{DSCS}$) reports the performance when only DSCS is applied: gradient ascent is applied to domain-specific channels, but using the entire training set. This setting also results in degraded performance, as globally unlearning domain-specific features leads to excessive simplification, thereby diminishing the model's representational capacity and impairing generalization to unseen domains. In contrast, the fourth row ($ERM_{IU}$) demonstrates that combining USS and DSCS yields the best performance. These two components are complementary: removing one while retaining the other leads to suboptimal outcomes and compromises model effectiveness. We further compare Inter-Domain Variance with Aggregated Variance and evaluate the proposed Domain-Specific Gradient Ascent against other unlearning strategies. Detailed results and analyses are provided in Appendix E.

## 4.4 FURTHER ANALYSIS

**Change in Unlearning Score w/o EMA**. Figure 3 shows the evolution of unlearning scores over training epochs (1 to 50) for six randomly selected training samples, comparing results with and without Exponential Moving Average (EMA) smoothing. As shown, the unlearning scores in both settings eventually converge. This convergence reflects the stability of each sample's influence on the model and offers insights into its learnability and memorization behavior. As illustrated in Figure 3b, EMA smoothing significantly reduces noise and yields smoother score trajectories. Notably, it also improves the separability between samples; for instance, Data 2 and Data 3 begin to diverge earlier, offering clearer distinctions in their contributions to model behavior.

Table 1: Leave-one-domain-out results on benchmarks (with 95% confidence intervals). OH denotes OfficeHome, Terra denotes Terra Incognita, DN denotes DomainNet, and NICO denotes NICO++.

| | PACS | OH | VLCS | Terra | DN | Digits | NICO |
|---|---|---|---|---|---|---|---|
| ERM (1999) | $83.0 \pm .4$ | $68.2 \pm .6$ | $77.2 \pm .5$ | $41.7 \pm .6$ | $40.7 \pm .4$ | $79.4 \pm .3$ | $79.8 \pm .3$ |
| ERM$_{IU}$ | $85.7 \pm .3$ | $69.8 \pm .6$ | $80.0 \pm .4$ | $44.2 \pm .3$ | $42.2 \pm .5$ | $82.1 \pm .4$ | $81.2 \pm .5$ |
| ERM$_{IUE}$ | $\mathbf{86.0 \pm .2}$ | $\mathbf{70.0 \pm .5}$ | $\mathbf{80.6 \pm .5}$ | $\mathbf{44.5 \pm .5}$ | $\mathbf{43.1 \pm .3}$ | $\mathbf{82.9 \pm .4}$ | $\mathbf{81.5 \pm .4}$ |
| MMD (2018b) | $83.2 \pm .7$ | $67.7 \pm .6$ | $77.2 \pm .4$ | $46.6 \pm .6$ | $31.7 \pm .5$ | $79.9 \pm .4$ | $80.2 \pm .4$ |
| MMD$_{IU}$ | $84.6 \pm .4$ | $69.8 \pm .5$ | $80.1 \pm .4$ | $48.4 \pm .4$ | $34.2 \pm .6$ | $81.2 \pm .4$ | $82.5 \pm .5$ |
| MMD$_{IUE}$ | $\mathbf{84.9 \pm .4}$ | $\mathbf{70.4 \pm .4}$ | $\mathbf{80.7 \pm .4}$ | $\mathbf{48.9 \pm .5}$ | $\mathbf{34.6 \pm .6}$ | $\mathbf{81.9 \pm .4}$ | $\mathbf{83.0 \pm .5}$ |
| IRM (2019) | $81.5 \pm .3$ | $66.9 \pm .4$ | $76.4 \pm .4$ | $43.1 \pm .6$ | $36.0 \pm .4$ | $79.2 \pm .4$ | $79.3 \pm .4$ |
| IRM$_{IU}$ | $82.8 \pm .2$ | $69.7 \pm .6$ | $77.9 \pm .4$ | $45.4 \pm .7$ | $38.1 \pm .3$ | $80.3 \pm .5$ | $81.9 \pm .2$ |
| IRM$_{IUE}$ | $\mathbf{83.2 \pm .2}$ | $\mathbf{70.3 \pm .7}$ | $\mathbf{78.3 \pm .4}$ | $\mathbf{46.0 \pm .7}$ | $\mathbf{38.8 \pm .2}$ | $\mathbf{80.7 \pm .6}$ | $\mathbf{82.3 \pm .2}$ |
| DDAIG (2020) | $83.2 \pm .3$ | $69.9 \pm .3$ | $76.7 \pm .3$ | $45.2 \pm .2$ | $41.5 \pm .3$ | $80.2 \pm .3$ | $81.4 \pm .2$ |
| DDAIG$_{IU}$ | $85.1 \pm .4$ | $71.6 \pm .2$ | $78.8 \pm .4$ | $46.6 \pm .3$ | $42.9 \pm .2$ | $81.7 \pm .3$ | $82.6 \pm .3$ |
| DDAIG$_{IUE}$ | $\mathbf{85.5 \pm .3}$ | $\mathbf{71.8 \pm .3}$ | $\mathbf{79.6 \pm .4}$ | $\mathbf{47.3 \pm .2}$ | $\mathbf{43.6 \pm .2}$ | $\mathbf{81.9 \pm .2}$ | $\mathbf{82.7 \pm .2}$ |
| MixStyle (2021) | $82.3 \pm .3$ | $70.5 \pm .3$ | $77.5 \pm .3$ | $49.0 \pm .3$ | $42.8 \pm .3$ | $81.4 \pm .3$ | $82.3 \pm .3$ |
| MixStyle$_{IU}$ | $84.2 \pm .2$ | $72.0 \pm .2$ | $79.2 \pm .4$ | $50.2 \pm .4$ | $43.9 \pm .4$ | $83.0 \pm .3$ | $83.5 \pm .4$ |
| MixStyle$_{IUE}$ | $\mathbf{85.1 \pm .2}$ | $\mathbf{72.9 \pm .2}$ | $\mathbf{79.4 \pm .4}$ | $\mathbf{50.9 \pm .4}$ | $\mathbf{44.6 \pm .3}$ | $\mathbf{83.6 \pm .2}$ | $\mathbf{84.1 \pm .3}$ |
| DomainMix (2022) | $82.2 \pm .3$ | $69.8 \pm .3$ | $76.1 \pm .3$ | $48.1 \pm .3$ | $42.3 \pm .2$ | $80.0 \pm .4$ | $82.7 \pm .3$ |
| DomainMix$_{IU}$ | $84.0 \pm .4$ | $71.2 \pm .4$ | $77.6 \pm .3$ | $50.4 \pm .4$ | $43.2 \pm .3$ | $81.4 \pm .5$ | $84.0 \pm .3$ |
| DomainMix$_{IUE}$ | $\mathbf{84.3 \pm .3}$ | $\mathbf{71.6 \pm .4}$ | $\mathbf{77.9 \pm .4}$ | $\mathbf{50.6 \pm .4}$ | $\mathbf{43.6 \pm .2}$ | $\mathbf{82.2 \pm .4}$ | $\mathbf{85.0 \pm .4}$ |
| EFDMix (2022) | $84.6 \pm .4$ | $71.2 \pm .2$ | $78.3 \pm .3$ | $49.9 \pm .3$ | $44.2 \pm .3$ | $82.1 \pm .3$ | $82.6 \pm .3$ |
| EFDMix$_{IU}$ | $86.1 \pm .5$ | $72.3 \pm .3$ | $79.4 \pm .4$ | $51.1 \pm .2$ | $45.3 \pm .2$ | $83.7 \pm .2$ | $83.9 \pm .4$ |
| EFDMix$_{IUE}$ | $\mathbf{86.6 \pm .4}$ | $\mathbf{73.1 \pm .3}$ | $\mathbf{80.1 \pm .3}$ | $\mathbf{51.5 \pm .3}$ | $\mathbf{45.6 \pm .2}$ | $\mathbf{84.3 \pm .2}$ | $\mathbf{84.8 \pm .3}$ |
| RISE (2023) | $86.3 \pm .4$ | $71.1 \pm .2$ | $80.6 \pm .3$ | $34.4 \pm .3$ | $45.4 \pm .2$ | $51.6 \pm .3$ | $82.9 \pm .4$ |
| RISE$_{IU}$ | $87.4 \pm .3$ | $72.4 \pm .3$ | $82.1 \pm .3$ | $35.7 \pm .2$ | $46.6 \pm .2$ | $53.0 \pm .4$ | $84.6 \pm .5$ |
| RISE$_{IUE}$ | $\mathbf{88.3 \pm .2}$ | $\mathbf{73.0 \pm .4}$ | $\mathbf{82.5 \pm .3}$ | $\mathbf{36.1 \pm .2}$ | $\mathbf{46.8 \pm .3}$ | $\mathbf{53.5 \pm .3}$ | $\mathbf{85.3 \pm .5}$ |
| SSPL (2024) | $84.0 \pm .3$ | $71.2 \pm .2$ | $77.9 \pm .4$ | $48.5 \pm .3$ | $42.8 \pm .3$ | $81.1 \pm .3$ | $82.3 \pm .3$ |
| SSPL$_{IU}$ | $85.6 \pm .2$ | $72.7 \pm .2$ | $80.3 \pm .3$ | $50.9 \pm .4$ | $44.2 \pm .2$ | $83.3 \pm .3$ | $84.0 \pm .2$ |
| SSPL$_{IUE}$ | $\mathbf{85.8 \pm .2}$ | $\mathbf{73.4 \pm .2}$ | $\mathbf{81.3 \pm .3}$ | $\mathbf{51.2 \pm .3}$ | $\mathbf{45.1 \pm .2}$ | $\mathbf{83.6 \pm .2}$ | $\mathbf{84.8 \pm .2}$ |
| LFME (2024) | $85.2 \pm .4$ | $69.4 \pm .3$ | $78.9 \pm .3$ | $48.1 \pm .7$ | $41.8 \pm .3$ | $81.9 \pm .2$ | $82.8 \pm .3$ |
| LFME$_{IU}$ | $87.3 \pm .4$ | $71.4 \pm .3$ | $80.2 \pm .4$ | $50.2 \pm .7$ | $42.7 \pm .4$ | $83.3 \pm .3$ | $84.2 \pm .3$ |
| LFME$_{IUE}$ | $\mathbf{87.9 \pm .3}$ | $\mathbf{71.9 \pm .2}$ | $\mathbf{81.1 \pm .3}$ | $\mathbf{50.9 \pm .7}$ | $\mathbf{43.1 \pm .2}$ | $\mathbf{83.5 \pm .2}$ | $\mathbf{84.6 \pm .2}$ |
| UDIM (2024) | $84.7 \pm .3$ | $68.9 \pm .4$ | $77.5 \pm .4$ | $47.3 \pm .6$ | $40.5 \pm .4$ | $80.8 \pm .3$ | $81.9 \pm .4$ |
| UDIM$_{IU}$ | $86.6 \pm .3$ | $70.1 \pm .3$ | $80.2 \pm .3$ | $50.9 \pm .7$ | $44.3 \pm .2$ | $83.5 \pm .3$ | $84.4 \pm .3$ |
| UDIM$_{IUE}$ | $\mathbf{87.3 \pm .4}$ | $\mathbf{70.4 \pm .3}$ | $\mathbf{80.6 \pm .4}$ | $\mathbf{51.2 \pm .7}$ | $\mathbf{44.7 \pm .3}$ | $\mathbf{84.4 \pm .3}$ | $\mathbf{84.6 \pm .3}$ |
| VL2V (2024) | $85.5 \pm .5$ | $69.8 \pm .3$ | $79.2 \pm .2$ | $48.6 \pm .8$ | $42.1 \pm .3$ | $82.3 \pm .2$ | $83.2 \pm .3$ |
| VL2V$_{IU}$ | $87.0 \pm .4$ | $71.6 \pm .4$ | $80.5 \pm .2$ | $50.2 \pm .7$ | $44.3 \pm .3$ | $84.4 \pm .3$ | $84.5 \pm .4$ |
| VL2V$_{IUE}$ | $\mathbf{87.9 \pm .3}$ | $\mathbf{72.3 \pm .3}$ | $\mathbf{80.7 \pm .2}$ | $\mathbf{50.5 \pm .7}$ | $\mathbf{44.8 \pm .2}$ | $\mathbf{84.7 \pm .2}$ | $\mathbf{85.3 \pm .6}$ |

Table 2: Ablation Study for Unlearning Set Selection and Domain-Specific Channel Selection.

| | PACS | OH | VLCS | Terra | DN | Digits | NICO |
|---|---|---|---|---|---|---|---|
| ERM | $83.0 \pm .4$ | $68.2 \pm .6$ | $77.2 \pm .5$ | $41.7 \pm .6$ | $40.7 \pm .4$ | $79.4 \pm .3$ | $79.8 \pm .3$ |
| ERM$_{USS}$ | $78.9 \pm .5$ | $64.3 \pm .4$ | $72.6 \pm .6$ | $37.6 \pm .6$ | $37.4 \pm .6$ | $74.5 \pm .2$ | $76.3 \pm .3$ |
| ERM$_{DSCS}$ | $76.7 \pm .6$ | $62.5 \pm .5$ | $70.4 \pm .5$ | $36.3 \pm .4$ | $34.6 \pm .5$ | $72.9 \pm .3$ | $74.6 \pm .5$ |
| ERM$_{IU}$ | $85.7 \pm .3$ | $69.8 \pm .6$ | $80.0 \pm .4$ | $44.2 \pm .3$ | $42.2 \pm .5$ | $82.1 \pm .4$ | $81.2 \pm .5$ |

**Distribution of Unlearning Score w/o EMA**. As shown in Figure 4, applying EMA smoothing (Figure 4b) results in a broader distribution of unlearning scores compared to the non-EMA setting (Figure 4a). This increased spread improves the signal-to-noise ratio, enhancing the separation between samples with high and low unlearning influence. The resulting distribution more clearly highlights influential outliers, enabling more accurate and confident selection of the unlearning set.

**Distribution of IDV**. Figure 5 illustrates that IDV provides a strong discriminative signal for identifying domain-specific channels. The majority of channels exhibit low IDV values and are thus regarded as domain-invariant, while a small subset forms a long-tail distribution with substantially higher IDV values, corresponding to domain-specific channels. This clear bimodal pattern, charac-

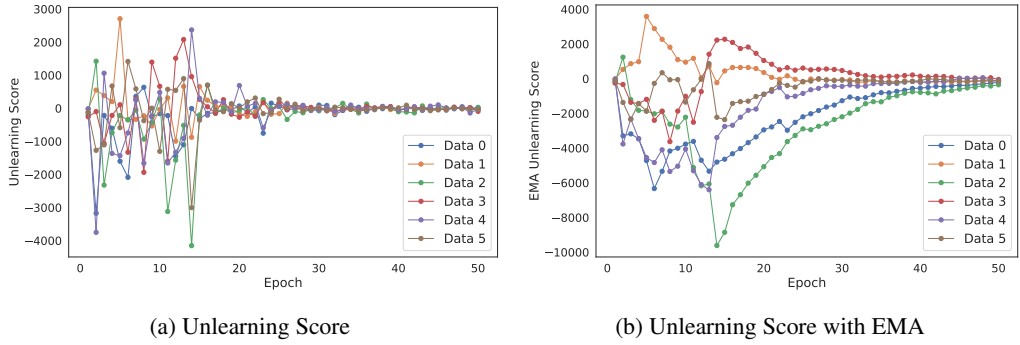

(a) Unlearning Score

(b) Unlearning Score with EMA

Figure 3: Comparison of unlearning scores with and without Exponential Moving Average (EMA).

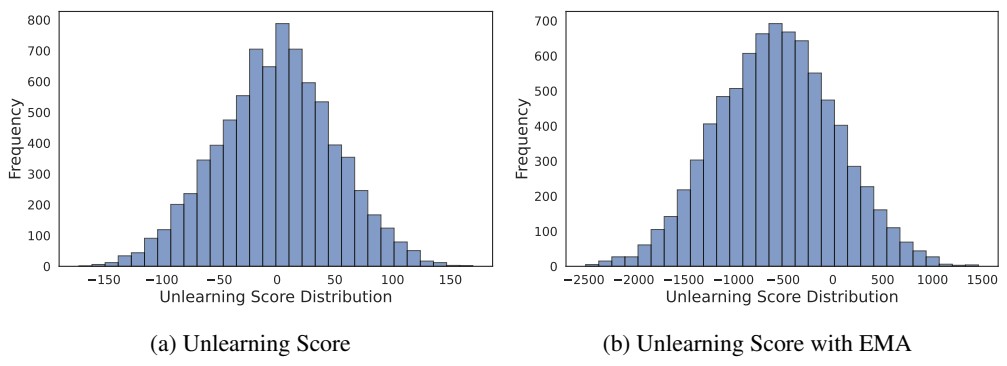

(a) Unlearning Score

(b) Unlearning Score with EMA

Figure 4: Unlearning Score w/o Exponential Moving Average

terized by a dense concentration of low-variance channels and a sparse tail of high-variance ones, demonstrates the effectiveness of IDV in separating domain-invariant and domain-specific channels.

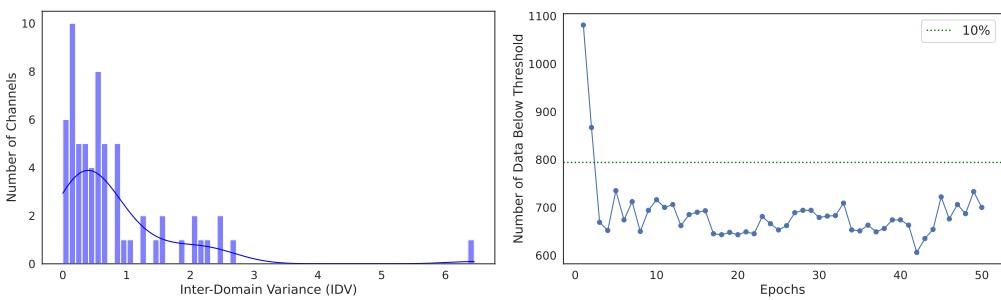

Figure 5: Distribution of IDV

Figure 6: Number of Data Unlearned

**Number of Samples Selected During Training**. Figure 6 shows the number of training samples selected for unlearning during training in the PACS dataset. Fixed thresholding strategies, such as selecting the bottom 10% of samples based on unlearning scores (green dashed line), are inflexible and fail to adapt to the evolving distribution of scores during training. In contrast, dynamic thresholding methods like MAD naturally adapt to score fluctuations, particularly in the early, unstable training phases. This adaptability improves both the reliability and precision of unlearning set selection, reducing the likelihood of over-selection or under-selection.

**Others**. We further provide supporting evidence through T-SNE visualizations, model complexity analyses, and computational cost evaluations. Details are presented in Appendices G and H.

## 5 CONCLUSIONS

This paper introduced Identify and Unlearn (IU), a model-agnostic module designed to continually mitigate domain-specific reliance in deep neural networks. IU leverages the unlearning score to identify the unlearning set, the Inter-Domain Variance (IDV) metric to detect domain-specific channels, and a Domain-Specific Gradient-Ascent (DSGA) procedure to selectively remove domain-specific features while preserving domain-invariant ones. Extensive experiments demonstrate the effectiveness of adaptive, post-epoch unlearning as a principled complement to existing DG approaches. Looking forward, future work will explore integrating IU with large models, such as large multimodal models, to evaluate its scalability and transferability.

## ACKNOWLEDGEMENTS

This research was partially supported by the New Zealand MBIE Strategic Science Investment Fund (SSIF) Data Science platform – Time-Evolving Data Science / Artificial Intelligence for Advanced Open Environmental Science (UOWX1910). We also gratefully acknowledge the Centre for Machine Learning for Social Good and the Advanced Machine Learning and Data Analytics Research (MARS) Lab at the University of Auckland.

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

# APPENDICES

## A    ETHICS STATEMENT

This work adheres to the ICLR Code of Ethics. The study does not involve human subjects, personally identifiable information, or sensitive data, and all datasets used are publicly available under appropriate licenses. No private or proprietary information was collected, and no experiments pose risks of harm to individuals, groups, or the environment. The methods and findings are intended solely for advancing scientific understanding of domain generalization and are not designed for harmful applications. We have taken care to ensure fairness and reproducibility by documenting our approach and following established research integrity standards.

## B    PRELIMINARY

**Notation.** Let $\mathcal{X}$ denote an input feature space, with dimension $d$, and $\mathcal{Y}$ a target class label space. A domain, $\mathcal{D}$, is composed of data sampled from a joint distribution $\mathbb{P}(X, Y)$ on $\mathcal{X} \times \mathcal{Y}$, where $\mathcal{D} = (x_i, y_i)_{i=1}^n \sim \mathbb{P}(X, Y)$, $x \in \mathcal{X} \subset \mathbb{R}^d$, $y \in \mathcal{Y} \subset \mathbb{R}$ and $n$ is the number of data in the domain. Here, $X$ and $Y$ denote the corresponding random variables (Wang et al., 2022; Zhou et al., 2022).

**Domain Generalization.** For the task of domain generalization, the input is $N$ source domains (training set), $\mathcal{S} = \{\mathcal{D}^j \mid j = 1, \cdots, N\}$, where $\mathcal{D}^j = \{(x_i^j, y_i^j)\}_{i=1}^{n_j}$ denotes the $j$-th domain and $n_j$ denotes the number of examples in the $j$-th domain. The joint distributions between each pair of domains are different: $\mathbb{P}(X, Y)^{(j)} \neq \mathbb{P}(X, Y)^{(k)}, j \neq k$. The goal of domain generalization is to learn a robust and generalizable predictive function $f : \mathcal{X} \to \mathcal{Y}$ from the $N$ source domains to achieve a minimum prediction error on an unseen target domain $\mathcal{T}$, where $\mathcal{T}$ cannot be accessed during training and $\mathbb{P}(X, Y)^{(\mathcal{T})} \neq \mathbb{P}(X, Y)^{(j)}$ for $j \in \{1, \cdots, N\}$.

**Machine Unlearning.** Consider a subset of training samples designated for removal, termed the unlearning set $D_u$. A machine unlearning procedure $U(A(D), D, D_u)$ is formally defined as a mapping from a trained model $A(D)$, the original training dataset $D$, and the unlearning set $D_u$, to an updated model $w_u$. Here, $A(\cdot)$ represents an arbitrary machine learning algorithm.

## C    REPRODUCIBILITY STATEMENT

**Datasets.** We evaluate our module on the DomainBed benchmark (Gulrajani & Lopez-Paz, 2021), using five widely adopted datasets: PACS (Li et al., 2017), OfficeHome (Venkateswara et al., 2017), VLCS (Fang et al., 2013), Terra Incognita (Beery et al., 2018), and DomainNet (Peng et al., 2019). In addition, we assess generalization performance on Digits-DG, following the protocol of (Zhou et al., 2020), and on NICO++, following the setup in (Zhang et al., 2023).

**Baselines.** We evaluate the effectiveness and model-agnostic nature of our unlearning module by integrating it with a diverse set of state-of-the-art domain generalization methods across multiple categories. Specifically, we consider CrossGrad (Shankar et al., 2018), MLDG (Li et al., 2018a), MMD (Li et al., 2018b), IRM (Arjovsky et al., 2019), DDAIG (Zhou et al., 2020), RSC (Huang et al., 2020), MixStyle (Zhou et al., 2021), DomainMix (Sun et al., 2022), EFDMix (Zhang et al., 2022), RISE (Huang et al., 2023), DomainDrop (Guo et al., 2023), SSPL (Zhao et al., 2024), LFME (Chen et al., 2024), UDIM (Shin et al., 2024), and VL2V (Addepalli et al., 2024). In addition, we include a standard Empirical Risk Minimization (ERM) baseline (Vapnik, 1999), which trains on the combined data from all source domains without applying any domain generalization techniques.

**Evaluation Metrics.** Following prior works (Gulrajani & Lopez-Paz, 2021; Zhou et al., 2022), we adopt a leave-one-domain-out evaluation protocol. In each run, one domain is held out as the target domain for testing, while the remaining domains serve as source domains for training. We report the top-1 classification accuracy (%) averaged over ten runs, along with the corresponding 95% confidence intervals.

**Implementation Details.** All input images across benchmarks are resized to $224 \times 224$ pixels. In our experiments, aligned with standard practices in the domain generalization literature, the validation set is constructed by uniformly sampling from all source domains, ensuring it is representative of the training distribution. Since we do not assume access to any target domain data, the validation set does not introduce target-domain bias. For non-CLIP-based methods, we adopt a pretrained ResNet-50 as the backbone. For CLIP-based methods, both ResNet-50 and ViT-B/32 are evaluated, with ViT-B/32 results reported in the supplementary material. All implementations are developed using the PyTorch library. Models are trained using Stochastic Gradient Descent (SGD) with a momentum of 0.9 and a weight decay of $5 \times 10^{-4}$. Training is conducted for 50 epochs with a batch size of 64 and an initial learning rate of 0.001, scheduled via the CosineAnnealingLR scheduler. All experiments are run on NVIDIA Tesla A100 GPUs (80GB) with CUDA version 12.1. The balancing parameter $\alpha$ is set to 1.1.

## D    ALGORITHM

## E    ADDITIONAL ABLATION STUDY

**Comparing IDV with Aggregated Variance.** Table 3 presents evaluation results comparing Domain-Specific Gradient Ascent (DSGA) under different channel selection strategies. Here, AV refers to Aggregated Variance, and IDV refers to Inter-Domain Variance. We draw two key observations: (1) Even when domain-specific channels are identified using Aggregated Variance, IU still yields substantial performance improvements over the original baseline, demonstrating the effectiveness of the proposed unlearning strategy. (2) When Inter-Domain Variance is used for channel selection, performance gains are further enhanced, highlighting the advantage of IDV over Aggregated Variance in accurately identifying domain-specific channels.

Table 3: Comparing IDV with Aggregated Variance.

|  | PACS | OH | VLCS | Terra | DN | Digits | NICO |
|---|---|---|---|---|---|---|---|
| ERM | $83.0 \pm .4$ | $68.2 \pm .6$ | $77.2 \pm .5$ | $41.7 \pm .6$ | $40.7 \pm .4$ | $79.4 \pm .3$ | $79.8 \pm .3$ |
| ERM$_{AV}$ | $84.3 \pm .4$ | $69.1 \pm .5$ | $78.5 \pm .5$ | $43.2 \pm .6$ | $42.0 \pm .6$ | $81.2 \pm .3$ | $80.8 \pm .3$ |
| ERM$_{IDV}$ | $\mathbf{85.7 \pm .3}$ | $\mathbf{69.8 \pm .6}$ | $\mathbf{80.0 \pm .4}$ | $\mathbf{44.2 \pm .3}$ | $\mathbf{42.2 \pm .5}$ | $\mathbf{82.1 \pm .4}$ | $\mathbf{81.2 \pm .5}$ |

**Comparing DSGA with Other Unlearning Methods.** We do not compare Domain-Specific Gradient Ascent (DSGA) against existing unlearning methods, as, to the best of our knowledge, no un-

---

**Algorithm 1** Identify and Unlearn (IU)

---

1: **Input:** $D$: training set; $D_{\text{val}}$: validation set; $\mathcal{C}$: set of channels; $A(\cdot)$: arbitrary DG method; $k$: MAD threshold multiplier; $\alpha$: parameter for unlearning score.
2: **Output:** $f_{\theta*}$: unlearned model with reduced domain-specific features.
3: **for** each epoch **do**
4:     $f_\theta = A(D)$                                        $\triangleright$ Train model with algorithm $A$ on training set $D$.
5:     **for** each training sample $x$ in $D$ **do**
6:         $C_x = \| -H_\theta^{-1} \nabla_\theta L(x, \theta) \|_2$                     $\triangleright$ Compute complexity score.
7:         $G_x = \sum_{z \in D_{\text{val}}} -\nabla_\theta L(z, \theta)^\top H_\theta^{-1} \nabla_\theta L(x, \theta)$     $\triangleright$ Compute generalization score.
8:         $U_x = \frac{(G_x)^\alpha}{C_x}$                                   $\triangleright$ Compute unlearning score.
9:     **end for**
10:    $\tau_u = U_{\tilde{x}} - k \times \text{median}(|U_{x_i} - U_{\tilde{x}}|, x_i \in D_{\text{train}})$ $\triangleright$ Compute threshold $\tau_u$ for selecting unlearning set.
11:    $D_u = \{x \in D \mid U_x < \tau_u\}$                      $\triangleright$ Select unlearning set $D_u$.
12:    **for** each channel $c$ **do**
13:       $\text{IDV}(c) = \text{Variance}\left(\left\{v_{(d)}^c\right\}_{d=1}^N\right), v_{(d)}^c = \frac{1}{N_d}\sum_{i=1}^{N_d}\left(x_{(d,i)}^c - \mu_{(d)}^c\right)^2$ $\triangleright$ Compute Inter-Domain Variance for channel $c$.
14:    **end for**
15:    $\tau_c = \text{IDV}_{\tilde{c}} - k \times \text{median}(|\text{IDV}_{c_i} - \text{IDV}_{\tilde{c}}|, c_i \in \mathcal{C})$       $\triangleright$ Compute threshold $\tau_c$ for selecting domain-specific channels.
16:    $\mathcal{C}_{\text{spc}} = \{c \mid \text{IDV}(c) > \tau_c\}$                $\triangleright$ Select domain-specific channels $\mathcal{C}_{\text{spc}}$.
17:    **for** each $x \in D_u$ **do**
18:       **for** each $c \in \mathcal{C}_{\text{spc}}$ **do**
19:         $\theta^c = \theta^c + \nabla_{\theta^c} L(x, \theta)$              $\triangleright$ Perform Domain-Specific Gradient Ascent.
20:       **end for**
21:    **end for**
22: **end for**
23: **Return** $f_{\theta*}$

---

learning strategy has been specifically designed for domain generalization. Unlearning techniques developed for generative models are not applicable in this context due to fundamental differences in model architectures and objectives. Similarly, model-agnostic unlearning approaches such as standard Gradient Ascent (Thudi et al., 2022) are primarily designed for privacy preservation rather than performance improvement. These methods often degrade model performance, and their upper bound is typically equivalent to a retrained model. In contrast, our approach is explicitly designed to enhance model generalization by selectively unlearning features that impair performance on unseen domains. Finally, unlearning strategies tailored for domain adaptation are also inapplicable here, as they require access to the target domain, which is unavailable in the domain generalization setting.

**Removing the unlearning set from the training data and retraining the model.** At specific training epochs, we remove the unlearning set entirely from the training data and continue training the model on the remaining samples. Table 4 presents the results for this experiment, where the unlearning set is removed at epochs 5, 10, 15, and 20.

Table 4: Removing the unlearning set from the training data and retraining the model.

|  | PACS | OH | VLCS | Terra | DN | Digits | NICO |
|---|---|---|---|---|---|---|---|
| ERM | $83.0 \pm .4$ | $68.2 \pm .6$ | $77.2 \pm .5$ | $41.7 \pm .6$ | $40.7 \pm .4$ | $79.4 \pm .3$ | $79.8 \pm .3$ |
| Epoch = 5 | $80.1 \pm .5$ | $66.0 \pm .6$ | $74.1 \pm .4$ | $39.1 \pm .5$ | $38.6 \pm .3$ | $76.4 \pm .3$ | $77.1 \pm .2$ |
| Epoch = 10 | $81.1 \pm .4$ | $66.8 \pm .5$ | $75.4 \pm .4$ | $40.2 \pm .4$ | $39.5 \pm .4$ | $77.8 \pm .4$ | $78.3 \pm .3$ |
| Epoch = 15 | $81.6 \pm .4$ | $67.1 \pm .5$ | $76.1 \pm .4$ | $40.8 \pm .4$ | $39.8 \pm .4$ | $78.3 \pm .4$ | $78.5 \pm .3$ |
| Epoch = 20 | $82.0 \pm .4$ | $67.4 \pm .5$ | $76.5 \pm .5$ | $41.1 \pm .5$ | $40.0 \pm .3$ | $78.7 \pm .3$ | $79.0 \pm .2$ |

As shown, this strategy consistently underperforms relative to the original ERM baseline. This is expected: completely removing the unlearning set is analogous to indiscriminately applying gradient ascent on all its associated features, both domain-invariant and domain-specific. This reduces the model's ability to learn useful information from these samples and decreases the overall training data size, ultimately leading to degraded performance.

Moreover, we observe that the earlier the unlearning set is removed, the more significant the performance drop. This trend is consistent with our analysis in Figure 6, which shows the unlearning set is

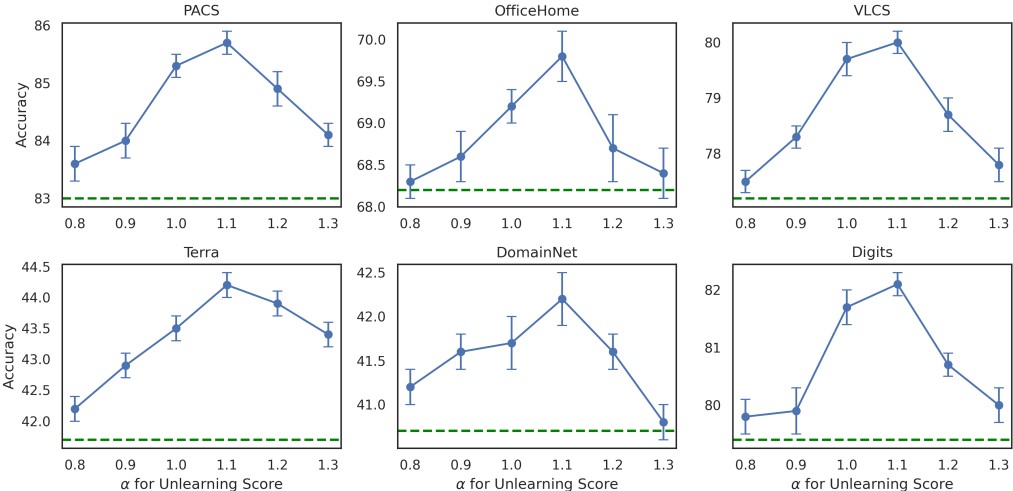

Figure 7: Sensitivity Analysis of $\alpha$ for Unlearning Score.

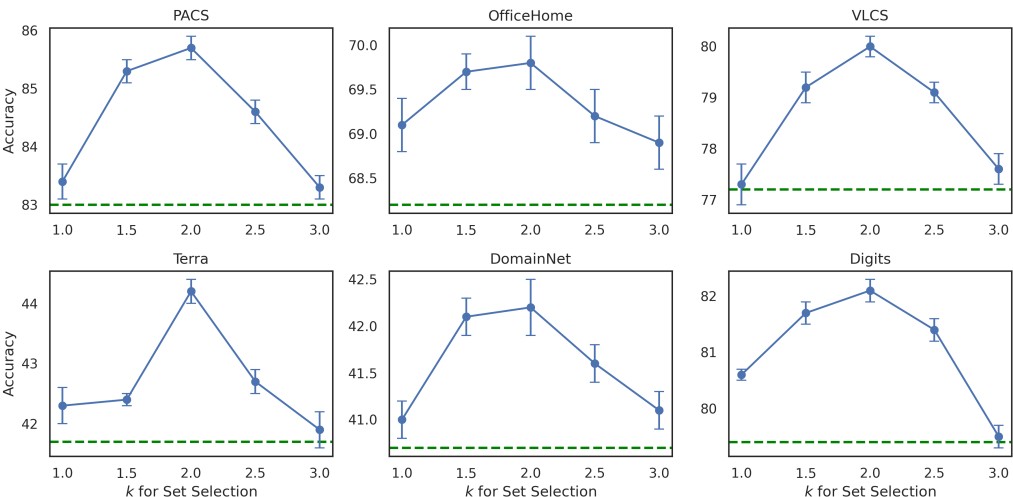

Figure 8: Sensitivity Analysis of $k$ for Unlearning Set Selection.

larger in the early stages of training. Thus, early removal eliminates more training data, amplifying the negative impact on generalization.

## F  SENSITIVITY ANALYSIS

Figures 7– 9 present the sensitivity analysis of $\alpha$ in the unlearning score; $k$ in MAD thresholding for unlearning set selection; and $k$ in MAD thresholding for domain-specific channel selection. As shown in Figure 7, except for extreme values of $\alpha$, IU consistently outperforms baselines, indicating insensitivity to $\alpha$. Figures 8 and 9 further show that performance gains remain substantial even with relatively small or large values of $k$, demonstrating that IU is similarly insensitive to the choice of $k$ in MAD thresholding. These results highlight the effectiveness of the proposed unlearning score and Inter-Domain Variance in reliably identifying both the unlearning set and domain-specific channels. Detailed sensitivity results are provided in Tables 5– 7.

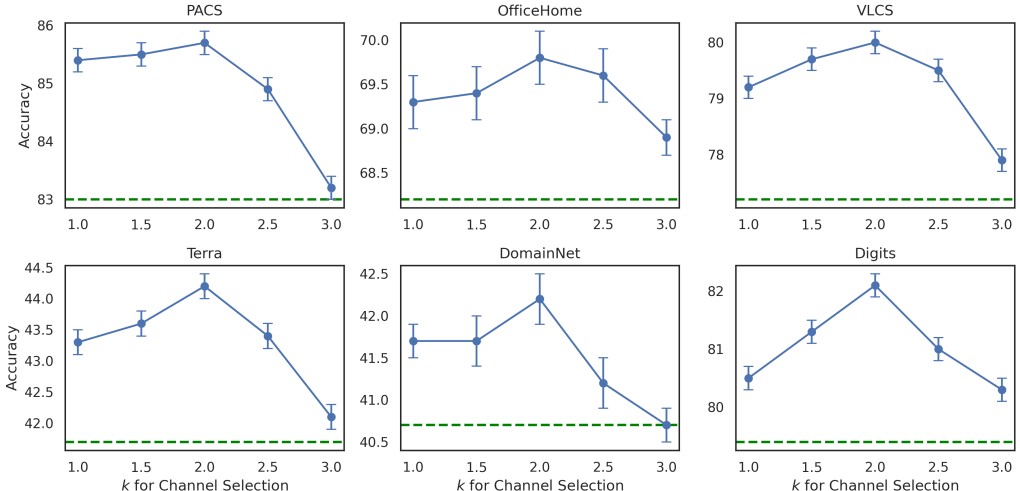

Figure 9: Sensitivity Analysis of $k$ for Channel Selection.

Table 5: Sensitivity Analysis of $\alpha$ for Unlearning Score.

|  | PACS | OH | VLCS | Terra | DN | Digits | NICO |
|---|---|---|---|---|---|---|---|
| ERM | $83.0 \pm .4$ | $68.2 \pm .6$ | $77.2 \pm .5$ | $41.7 \pm .6$ | $40.7 \pm .4$ | $79.4 \pm .3$ | $79.8 \pm .3$ |
| $\alpha = 0.8$ | $83.6 \pm .3$ | $68.3 \pm .2$ | $77.5 \pm .2$ | $42.2 \pm .2$ | $41.2 \pm .2$ | $79.8 \pm .3$ | $79.9 \pm .3$ |
| $\alpha = 0.9$ | $84.0 \pm .3$ | $68.6 \pm .3$ | $78.3 \pm .2$ | $42.9 \pm .2$ | $41.6 \pm .2$ | $79.9 \pm .4$ | $80.1 \pm .2$ |
| $\alpha = 1.0$ | $85.3 \pm .2$ | $69.2 \pm .2$ | $79.7 \pm .3$ | $43.5 \pm .2$ | $41.7 \pm .3$ | $81.7 \pm .3$ | $80.6 \pm .2$ |
| $\alpha = 1.1$ | $\mathbf{85.7 \pm .2}$ | $\mathbf{69.8 \pm .3}$ | $\mathbf{80.0 \pm .2}$ | $\mathbf{44.2 \pm .2}$ | $\mathbf{42.2 \pm .3}$ | $\mathbf{82.1 \pm .2}$ | $\mathbf{81.2 \pm .3}$ |
| $\alpha = 1.2$ | $84.9 \pm .3$ | $68.7 \pm .4$ | $78.7 \pm .3$ | $43.9 \pm .2$ | $41.6 \pm .2$ | $80.7 \pm .2$ | $80.2 \pm .3$ |
| $\alpha = 1.3$ | $84.1 \pm .2$ | $68.4 \pm .3$ | $77.8 \pm .3$ | $43.4 \pm .2$ | $40.8 \pm .2$ | $80.0 \pm .3$ | $80.0 \pm .3$ |

## G  FURTHER ANALYSIS

**T-SNE Visualization**. Figure 10 shows the T-SNE visualizations of embeddings produced by ERM and DomainDrop, with and without the IU module. For ERM (Figures 10a and 10b), incorporating IU yields a noticeably more separable embedding structure, suggesting that removing domain-specific features promotes learning domain-invariant representations and enhances generalization. For DomainDrop (Figures 10c and 10d), which already produces more separable embeddings than ERM, IU further improves cluster separation, demonstrating that IU complements existing DG methods rather than merely compensating for weak baselines.

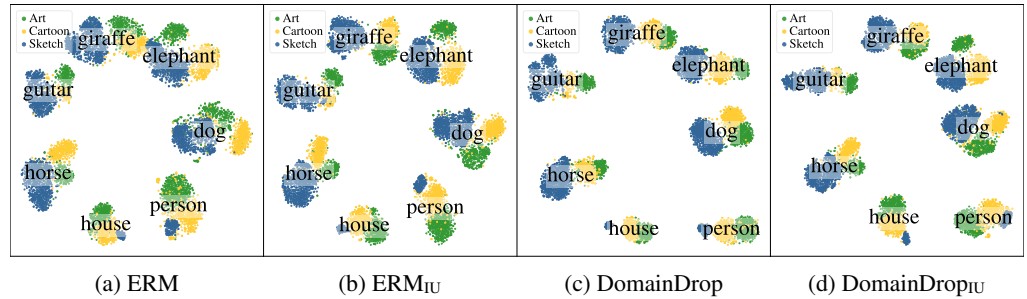

(a) ERM  (b) ERM$_{IU}$  (c) DomainDrop  (d) DomainDrop$_{IU}$

Figure 10: T-SNE Visualization for ERM and DomainDrop w/o IU

**Regularization versus Unlearning.** We further investigate whether our domain-specific channel selection method can serve as a form of regularization during training. In this regularization setting, we do not perform unlearning set identification or apply Domain-Specific Gradient Ascent. Instead, after identifying domain-specific channels, we apply regularization to features from those channels

Table 6: Sensitivity Analysis of $k$ for Unlearning Set Selection.

|  | PACS | OH | VLCS | Terra | DN | Digits | NICO |
|---|---|---|---|---|---|---|---|
| ERM | $83.0 \pm .4$ | $68.2 \pm .6$ | $77.2 \pm .5$ | $41.7 \pm .6$ | $40.7 \pm .4$ | $79.4 \pm .3$ | $79.8 \pm .3$ |
| $k = 1.0$ | $83.4 \pm .3$ | $69.1 \pm .3$ | $77.3 \pm .4$ | $42.3 \pm .3$ | $41.0 \pm .2$ | $80.6 \pm .1$ | $80.4 \pm .2$ |
| $k = 1.5$ | $85.3 \pm .2$ | $69.7 \pm .2$ | $79.2 \pm .3$ | $42.4 \pm .1$ | $42.1 \pm .2$ | $81.7 \pm .2$ | $80.6 \pm .3$ |
| $k = 2.0$ | $\mathbf{85.7 \pm .2}$ | $\mathbf{69.8 \pm .3}$ | $\mathbf{80.0 \pm .2}$ | $\mathbf{44.2 \pm .2}$ | $\mathbf{42.2 \pm .3}$ | $\mathbf{82.1 \pm .2}$ | $\mathbf{81.2 \pm .3}$ |
| $k = 2.5$ | $84.6 \pm .2$ | $69.2 \pm .3$ | $79.1 \pm .2$ | $42.7 \pm .2$ | $41.6 \pm .2$ | $81.4 \pm .2$ | $80.9 \pm .3$ |
| $k = 3.0$ | $83.3 \pm .2$ | $68.9 \pm .3$ | $77.6 \pm .3$ | $41.9 \pm .3$ | $41.1 \pm .2$ | $79.5 \pm .2$ | $80.3 \pm .2$ |

Table 7: Sensitivity Analysis of $k$ for Domain-Specific Channel Selection.

|  | PACS | OH | VLCS | Terra | DN | Digits | NICO |
|---|---|---|---|---|---|---|---|
| ERM | $83.0 \pm .4$ | $68.2 \pm .6$ | $77.2 \pm .5$ | $41.7 \pm .6$ | $40.7 \pm .4$ | $79.4 \pm .3$ | $79.8 \pm .3$ |
| $k = 1.0$ | $85.4 \pm .2$ | $69.3 \pm .3$ | $79.2 \pm .2$ | $43.3 \pm .2$ | $41.7 \pm .2$ | $80.5 \pm .2$ | $80.5 \pm .3$ |
| $k = 1.5$ | $85.5 \pm .2$ | $69.4 \pm .3$ | $79.7 \pm .2$ | $43.6 \pm .2$ | $41.7 \pm .3$ | $81.3 \pm .2$ | $80.8 \pm .3$ |
| $k = 2.0$ | $\mathbf{85.7 \pm .2}$ | $\mathbf{69.8 \pm .3}$ | $\mathbf{80.0 \pm .2}$ | $\mathbf{44.2 \pm .2}$ | $\mathbf{42.2 \pm .3}$ | $\mathbf{82.1 \pm .2}$ | $\mathbf{81.2 \pm .3}$ |
| $k = 2.5$ | $84.9 \pm .2$ | $69.6 \pm .3$ | $79.5 \pm .2$ | $43.4 \pm .2$ | $41.2 \pm .3$ | $81.0 \pm .2$ | $80.0 \pm .3$ |
| $k = 3.0$ | $83.2 \pm .2$ | $68.9 \pm .2$ | $77.9 \pm .2$ | $42.1 \pm .2$ | $40.7 \pm .2$ | $80.3 \pm .2$ | $79.9 \pm .3$ |

throughout training. As shown in Table 8, this regularization strategy also leads to performance improvements by suppressing the learning of domain-specific features. However, it remains less effective than our unlearning strategy. We attribute this performance gap to the difference in granularity: Unlearning leverages information across training epochs by identifying harmful samples and selectively applying gradient ascent between epochs, enabling a more global perspective. In contrast, regularization operates within mini-batches, providing only a local view, which can lead to suboptimal performance. This experiment highlights both the conceptual and practical differences between unlearning and regularization. Importantly, the two strategies are orthogonal and can be complementary. Our IU module is compatible with existing domain generalization methods, including those that employ regularization-based techniques.

Table 8: Evaluating Domain-Specific Channel Selection in the Regularization Setting.

|  | PACS | OH | VLCS | Terra | DN | Digits | NICO |
|---|---|---|---|---|---|---|---|
| ERM | $83.0 \pm .4$ | $68.2 \pm .6$ | $77.2 \pm .5$ | $41.7 \pm .6$ | $40.7 \pm .4$ | $79.4 \pm .3$ | $79.8 \pm .3$ |
| $ERM_{Reg}$ | $83.7 \pm .4$ | $69.3 \pm .4$ | $78.8 \pm .5$ | $42.1 \pm .6$ | $40.9 \pm .4$ | $80.2 \pm .3$ | $80.5 \pm .3$ |
| $ERM_{IU}$ | $\mathbf{85.7 \pm .3}$ | $\mathbf{69.8 \pm .6}$ | $\mathbf{80.0 \pm .4}$ | $\mathbf{44.2 \pm .3}$ | $\mathbf{42.2 \pm .5}$ | $\mathbf{82.1 \pm .4}$ | $\mathbf{81.2 \pm .5}$ |

**Model Complexity Change.** Figure 11 illustrates the reduction in model complexity (measured on ResNet-50) across training epochs for all benchmarks. The solid lines represent the unlearned model complexity, while the shaded regions indicate the magnitude of complexity reduction introduced by IU. We draw three key observations: (1) IU consistently reduces model complexity across all benchmarks, highlighting its effectiveness in simplifying the model by removing domain-specific knowledge. (2) The extent of complexity reduction is benchmark-dependent. For example, benchmarks such as VLCS and PACS exhibit relatively modest reductions, while larger gains are observed for Terra and NICO++, where the original complexity decreases more substantially over training. (3) The degree of complexity reduction correlates with the size of the benchmark. Larger datasets tend to result in a greater number of training samples selected for unlearning, thereby amplifying the impact of IU on model complexity.

## H    COMPUTATION COST ANALYSIS

Naive influence function estimation through direct inverse Hessian computation is prohibitively expensive for large-scale models. Our method circumvents this limitation by employing Hessian-Vector Products (HVP) combined with the LiSSA algorithm to efficiently approximate inverse-Hessian vector products. Below, we provide both theoretical and empirical analyses demonstrating the scalability of our approach when leveraging LiSSA for efficient approximation.

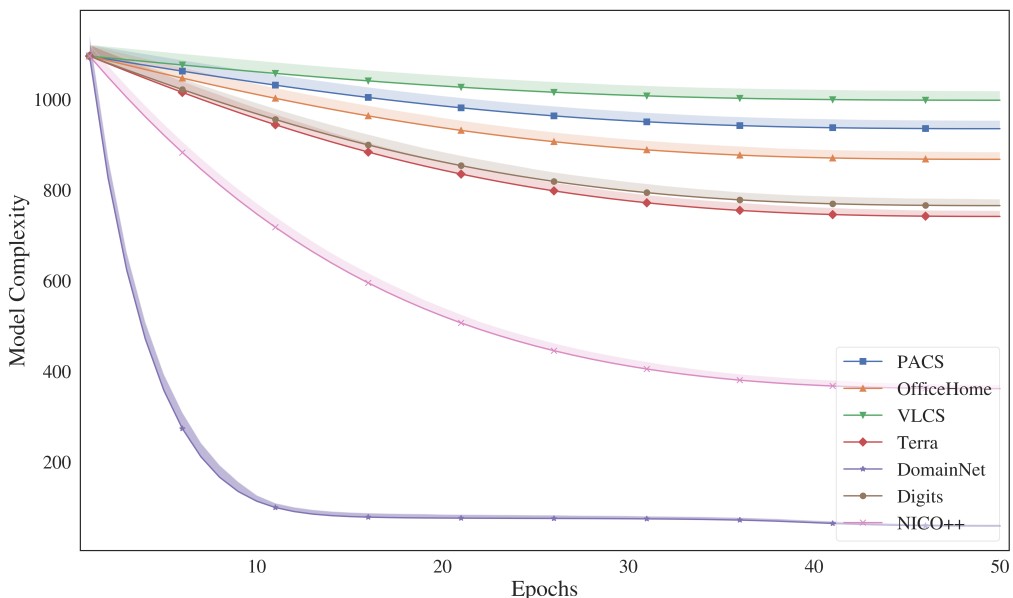

Figure 11: Model Complexity Change across Benchmarks.

**Theoretical Discussion for Scalability.** We compare three strategies (Table 9) for computing the inverse Hessian matrix: **Direct Hessian Inversion**, **Hessian-Vector Product + Conjugate Gradient**, and **Hessian-Vector Product + LiSSA**, a critical operation for influence function estimation. Each method differs in its computational efficiency, memory usage, and theoretical properties, particularly in the context of high-dimensional deep learning models.

Table 9: Comparing Three Strategies for Computing the Inverse Hessian Matrix.

| Method | Time | Memory | Explicit Hessian | Mini-batch |
|---|---|---|---|---|
| Direct Inversion | $\mathcal{O}(d^3)$ | $\mathcal{O}(d^2)$ | Yes | No |
| HVP + CG | $\mathcal{O}(T \cdot d)$ ($T$ = iterations) | $\mathcal{O}(d)$ | No (HVP only) | No |
| HVP + LiSSA | $\mathcal{O}(T \cdot d)$ ($T$ = recursion depth) | $\mathcal{O}(d)$ | No (HVP only) | Yes |

**Direct Hessian Inversion**. Direct Hessian Inversion requires computing and storing the full Hessian matrix $H \in \mathbb{R}^{d \times d}$, followed by matrix-vector multiplication. **Time complexity**: $\mathcal{O}(d^3)$ for matrix inversion. **Memory complexity**: $\mathcal{O}(d^2)$, required to store the full Hessian. **Theoretical properties**: Yields exact results when the full Hessian is available; however, infeasible for large-scale models due to cubic time and quadratic space complexity. **Limitation**: Not applicable in deep networks where $d$ can exceed millions.

**Hessian-Vector Product + Conjugate Gradient**. The inverse Hessian-vector product is reformulated as solving the linear system $Hx = v$, which is solved iteratively using Conjugate Gradient (CG). This avoids explicit Hessian storage and uses Hessian-vector products (HVPs), efficiently computed via reverse-mode automatic differentiation. **Time complexity**: $\mathcal{O}(T \cdot d)$, where $T$ is the number of CG iterations. **Memory complexity**: $\mathcal{O}(d)$, storing only vectors and intermediate results. **Theoretical properties**: Converges in at most $d$ iterations, faster convergence when $H$ is symmetric positive definite and well-conditioned.

**Hessian-Vector Product + LiSSA**. We leverage LiSSA (Agarwal et al., 2017) in our implementation, which provides a stochastic, iterative approximation to $H^{-1}v$ via a truncated Neumann series expansion. Unlike CG, LiSSA allows the use of stochastic Hessian estimates computed on mini-batches, improving scalability. **Time complexity**: $\mathcal{O}(T \cdot d)$, where $T$ is the number of recursion steps. **Memory complexity**: $\mathcal{O}(d)$, storing only vectors per iteration. **Theoretical properties**: Under strong convexity and smoothness assumptions, LiSSA exhibits geometric convergence. Convergence rate and stability improve with damping (i.e., computing $(H + \lambda I)^{-1}v$). **Advantage**:

Highly scalable and amenable to mini-batch computation; preferred in large-scale or online settings where CG is unstable or too costly.

**Empirical Discussion for Scalability.** Table 10 reports the per-epoch runtime of ERM and the three components of IU: domain-specific channel selection ($Set_{Channel}$), unlearning-set selection ($Set_{unlearn}$), and Domain-Specific Gradient Ascent (DSGA). All experiments use the settings and hardware described in the Appendix, with ResNet-18 as the backbone, and runtime measured in minutes. The cost of DSGA is marked as "–" because it simply adds the corresponding gradients back, and its overhead is negligible compared to the other steps.

Including ERM in the table clarifies the relative overhead of IU. As shown, $Set_{Channel}$ adds only a small cost over ERM, confirming that channel selection is lightweight. In contrast, the dominant cost arises from $Set_{unlearn}$, which performs influence-function-based scoring. The overhead scales with dataset size, reaching 354 minutes per epoch on the largest benchmark, DomainNet ($\approx 580k$ images). While non-trivial, this cost reflects the intrinsic difficulty of influence estimation at a large scale and remains feasible given the substantial performance gains IU delivers.

Importantly, this overhead does not need to occur every training epoch. As demonstrated in our accuracy–cost analysis (Table 11), invoking IU every $e \in \{2, 3, 4, 5\}$ epochs amortizes the running time by a factor of $e$ while retaining most of the accuracy improvement. For example, at $e = 2$, the effective cost is reduced by 50%, yet performance drops by less than 0.5% on average. Thus, IU can be operated in significantly more lightweight configurations without sacrificing its effectiveness.

Finally, although all measurements were conducted on a single A100 GPU, modern accelerators offer substantial additional speedups. Upgrading to an H200 reduces the $Set_{unlearn}$ cost on Domain-Net from 354 minutes to approximately 106 minutes per epoch. With data-parallel execution on 8×H200 GPUs, the cost can be further reduced to roughly 16 minutes per epoch, making IU increasingly practical with contemporary hardware.

Table 10: Running Time for IU (min)

|  | PACS | OH | VLCS | Terra | DN | Digits | NICO |
|---|---|---|---|---|---|---|---|
| ERM | $0.3 \pm 0.1$ | $0.5 \pm 0.1$ | $0.2 \pm 0.1$ | $0.9 \pm 0.2$ | $11.7 \pm 0.6$ | $0.9 \pm 0.1$ | $5.3 \pm 0.3$ |
| $Set_{Channel}$ | $0.4 \pm 0.1$ | $0.6 \pm 0.1$ | $0.2 \pm 0.1$ | $1.1 \pm 0.2$ | $29.5 \pm 0.6$ | $0.8 \pm 0.1$ | $6.1 \pm 0.2$ |
| $Set_{unlearn}$ | $39.6 \pm 5.8$ | $45.4 \pm 3.3$ | $9.8 \pm 1.3$ | $131.4 \pm 6.8$ | $354.5 \pm 9.4$ | $75.0 \pm 6.1$ | $184.6 \pm 9.9$ |
| DSGA | - | - | - | - | - | - | - |

## I   ACCURACY-COST TRADE-OFF ANALYSIS

To evaluate the computational efficiency of IU, we vary the frequency of invoking the module by performing unlearning every $e$ epochs ($e \in \{1, 2, 3, 4, 5\}$). Table 11 reports the results across seven benchmarks. We observe a clear and smooth trade-off between accuracy and cost. Executing IU every epoch ($e = 1$) yields the highest overall performance, but reducing the frequency leads to only gradual degradation. Notably, $e = 2$ retains more than 95% of the performance gain over ERM while reducing IU-related overhead by approximately 50%, making it a strong operating point in practice. Even with very sparse updates ($e \geq 4$), IU continues to outperform ERM on all benchmarks, demonstrating that the module does not rely on per-epoch execution to remain effective.

These results suggest that (i) the influence estimates and domain-specific channel scores evolve smoothly over training, so running IU intermittently is sufficient, and (ii) practitioners may select an appropriate update frequency depending on their computational budget. Overall, the findings confirm that IU provides consistent robustness improvements under a wide range of operating costs, enabling flexible deployment without sacrificing its core benefits.

## J   LONG-TERM STABILITY OF ITERATIVE INFLUENCE-BASED UNLEARNING

A natural question is whether repeatedly applying influence-based unlearning after each epoch could lead to the accumulation of approximation error and potentially cause the model to become unsta-

Table 11: Accuracy-Cost Trade-Off Analysis.

|  | PACS | OH | VLCS | Terra | DN | Digits | NICO |
|---|---|---|---|---|---|---|---|
| ERM | $83.0 \pm .4$ | $68.2 \pm .6$ | $77.2 \pm .5$ | $41.7 \pm .6$ | $40.7 \pm .4$ | $79.4 \pm .3$ | $79.8 \pm .3$ |
| $e = 1$ | $\mathbf{85.7 \pm .3}$ | $\mathbf{69.8 \pm .6}$ | $\mathbf{80.0 \pm .4}$ | $\mathbf{44.2 \pm .3}$ | $\mathbf{42.2 \pm .5}$ | $\mathbf{82.1 \pm .4}$ | $\mathbf{81.2 \pm .5}$ |
| $e = 2$ | $84.9 \pm .4$ | $69.3 \pm .7$ | $79.1 \pm .5$ | $43.5 \pm .4$ | $41.8 \pm .6$ | $81.5 \pm .4$ | $80.9 \pm .6$ |
| $e = 3$ | $84.3 \pm .5$ | $69.1 \pm .6$ | $78.6 \pm .6$ | $43.0 \pm .5$ | $41.6 \pm .5$ | $81.0 \pm .5$ | $80.7 \pm .5$ |
| $e = 4$ | $83.8 \pm .4$ | $68.7 \pm .7$ | $78.1 \pm .5$ | $42.6 \pm .4$ | $41.3 \pm .6$ | $80.4 \pm .4$ | $80.6 \pm .6$ |
| $e = 5$ | $83.4 \pm .5$ | $68.4 \pm .6$ | $77.9 \pm .6$ | $42.1 \pm .5$ | $41.0 \pm .5$ | $80.0 \pm .5$ | $80.1 \pm .5$ |

ble. Although IU relies on a first-order influence approximation, several design choices ensure that approximation error remains controlled throughout training.

**Localized interventions.** Domain-Specific Gradient Ascent (DSGA) applies gradient reversal only to a small subset of channels, those identified as domain-specific, and only using samples in the unlearning set. This produces highly localized updates, preventing large parameter drifts that could amplify approximation inaccuracies. Unlike full-parameter unlearning, IU's channel-level selectivity inherently constrains error propagation.

**Robust re-identification each epoch.** Rather than repeatedly acting on a fixed set of samples or channels (which could amplify early noise), IU re-identifies both the unlearning set and domain-specific channels at every invocation using robust MAD-based thresholds. This adaptivity prevents unstable samples or transient fluctuations from being reinforced across epochs, effectively damping potential accumulation of approximation error.

**Temporal smoothing via EMA.** We incorporate an exponential moving average (EMA) over unlearning scores to suppress short-term noise. EMA produces smoother trajectories for sample-level statistics, reducing sensitivity to momentary fluctuations and ensuring that IU reacts only to persistent, high-confidence domain-specific signals. This further mitigates the risk of instability caused by noisy influence estimates.

Collectively, these theoretical control mechanisms demonstrate that IU remains stable under iterative execution. While based on first-order approximations, IU's localized updates, robust adaptivity, and temporal smoothing prevent error accumulation and ensure reliable behavior throughout training.

# K   ADDITIONAL EXPERIMENTS RESULTS

Table 12: Leave-one-domain-out results on benchmarks (with 95% confidence intervals). OH denotes OfficeHome, Terra denotes Terra Incognita, DN denotes DomainNet, and NICO denotes NICO++.

|  | PACS | OH | VLCS | Terra | DN | Digits | NICO |
|---|---|---|---|---|---|---|---|
| CrossGrad (2018) | $81.7 \pm .3$ | $69.8 \pm .3$ | $76.1 \pm .3$ | $44.7 \pm .3$ | $38.5 \pm .2$ | $79.5 \pm .4$ | $80.6 \pm .3$ |
| CrossGrad$_{IU}$ | $84.2 \pm .2$ | $71.7 \pm .4$ | $79.0 \pm .5$ | $45.9 \pm .3$ | $40.7 \pm .1$ | $81.1 \pm .4$ | $82.3 \pm .4$ |
| CrossGrad$_{IUE}$ | $\mathbf{84.8 \pm .2}$ | $\mathbf{72.1 \pm .4}$ | $\mathbf{79.7 \pm .6}$ | $\mathbf{46.2 \pm .3}$ | $\mathbf{41.3 \pm .1}$ | $\mathbf{81.6 \pm .4}$ | $\mathbf{82.9 \pm .3}$ |
| MLDG (2018a) | $82.8 \pm .3$ | $68.6 \pm .4$ | $77.2 \pm .4$ | $46.2 \pm .5$ | $41.0 \pm .4$ | $79.7 \pm .4$ | $79.7 \pm .4$ |
| MLDG$_{IU}$ | $84.3 \pm .4$ | $70.8 \pm .3$ | $80.1 \pm .4$ | $48.0 \pm .7$ | $42.3 \pm .3$ | $81.4 \pm .5$ | $81.1 \pm .2$ |
| MLDG$_{IUE}$ | $\mathbf{84.9 \pm .4}$ | $\mathbf{71.2 \pm .4}$ | $\mathbf{80.7 \pm .4}$ | $\mathbf{48.4 \pm .6}$ | $\mathbf{43.0 \pm .3}$ | $\mathbf{81.7 \pm .6}$ | $\mathbf{81.6 \pm .1}$ |
| RSC (2020) | $82.7 \pm .5$ | $68.4 \pm .6$ | $77.5 \pm .4$ | $40.6 \pm .3$ | $39.0 \pm .4$ | $79.9 \pm .4$ | $82.1 \pm .4$ |
| RSC$_{IU}$ | $83.9 \pm .7$ | $70.5 \pm .6$ | $80.4 \pm .4$ | $42.3 \pm .5$ | $41.0 \pm .3$ | $81.3 \pm .5$ | $83.7 \pm .2$ |
| RSC$_{IUE}$ | $\mathbf{84.2 \pm .4}$ | $\mathbf{71.0 \pm .6}$ | $\mathbf{81.0 \pm .4}$ | $\mathbf{42.8 \pm .6}$ | $\mathbf{41.4 \pm .2}$ | $\mathbf{82.0 \pm .6}$ | $\mathbf{83.9 \pm .2}$ |
| DomainDrop (2023) | $89.3 \pm .3$ | $71.4 \pm .4$ | $79.1 \pm .4$ | $48.2 \pm .3$ | $43.7 \pm .4$ | $81.6 \pm .5$ | $82.6 \pm .4$ |
| DomainDrop$_{IU}$ | $90.5 \pm .3$ | $72.3 \pm .3$ | $80.5 \pm .5$ | $49.3 \pm .3$ | $45.0 \pm .4$ | $82.4 \pm .4$ | $84.1 \pm .5$ |
| DomainDrop$_{IUE}$ | $\mathbf{90.9 \pm .3}$ | $\mathbf{72.9 \pm .2}$ | $\mathbf{80.9 \pm .6}$ | $\mathbf{49.9 \pm .3}$ | $\mathbf{45.5 \pm .3}$ | $\mathbf{83.1 \pm .4}$ | $\mathbf{84.4 \pm .4}$ |

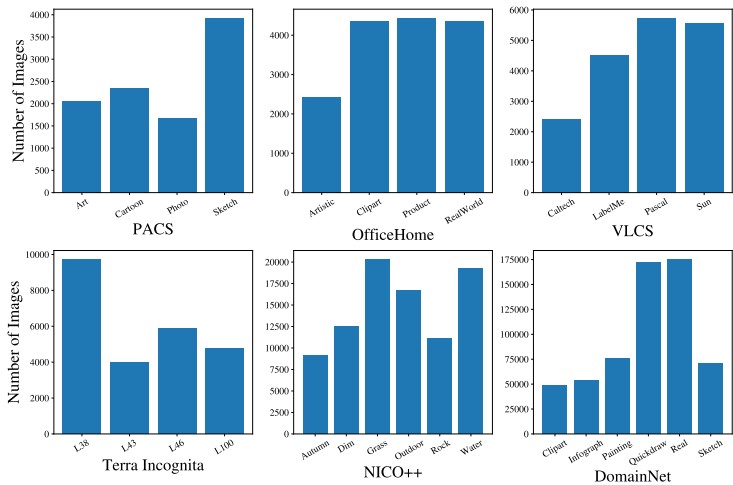

Figure 12: Dataset Distribution for PACS, OfficeHome, VLCS, Terra, NICO++, and DomainNet.

## L  ROBUSTNESS OF IDV UNDER DOMAIN IMBALANCE

All DG benchmarks used in this paper exhibit substantial domain-size imbalance (Figure 12). PACS, VLCS, TerraIncognita, NICO++, OfficeHome, and especially DomainNet show sample-size differences ranging from 2× to more than 4× across domains. This setting directly tests whether IDV is sensitive to such an imbalance. Methodologically, IDV is designed to be robust because it first computes within-domain variances and then measures the variance across domains, giving each domain equal weight regardless of its sample size. This avoids the dominant-domain bias inherent in Aggregated Variance (AV), where large domains can overshadow smaller ones. Empirically, IDV exhibits a stable, long-tailed distribution across all six benchmarks and consistently identifies a small subset of high-IDV channels, despite large domain-size discrepancies. Combined with the strong performance gains observed across benchmarks, these results suggest that IDV remains effective and stable in the face of severe domain imbalance.

