# OpenReview forum: "Unlearning during Training: Domain-Specific Gradient Ascent for Domain Generalization"
_ICLR.cc/2026/Conference — ICLR 2026 Poster_

### Official Review · Reviewer_CNti · 2025-10-31

**Soundness:** 3
**Presentation:** 2
**Contribution:** 2
**Rating:** 6
**Confidence:** 4

**Summary:**

This paper presents "Identify and Unlearn" (IU), a model-agnostic module that addresses domain generalization by continually mitigating reliance on domain-specific features during neural network training. IU combines an unlearning score (to identify training samples that disproportionately increase model complexity while failing to improve generalization), an Inter-Domain Variance (IDV) metric (to identify domain-specific channels), and a domain-specific gradient ascent procedure to selectively remove these harmful features while preserving domain-invariant content. The method is thoroughly evaluated on seven benchmarks and multiple baselines, demonstrating consistent improvements in out-of-distribution performance.

**Strengths:**

1.	The proposed inter-domain variance metric is a novel identification of domain-specific channels.
2.	The proposed UI module is model-agnostic, which could achieve performance improvements over multiple datasets and methods.

**Weaknesses:**

1.	The motivation is unclear. The authors claim that prior approaches (e.g., Aggregated Variance, AV) are highly sensitive to domain imbalance, whereas the proposed inter-domain variance is robust. However, this issue is neither revealed nor proved experimentally, as the datasets used do not appear to exhibit large domain imbalance. Besides, many DG methods already suppress domain-related features or channels during training, which seems to be in conflict with the existing methods that operate “without any mechanism to correct such reliance once it emerges.” Finally, the claim that AV ignores cross-domain variability also lacks direct evidence.
2.	The details of Fig. 1 should be provided. The experimental description of Fig. 1 is vague. The paper does not specify how the fur-edge channel and background-sensitive channel are distinguished. It is also unclear how the analyzed features are extracted.
3.	The contributions should be emphasized. The core contribution is the adaptive post-epoch removal of domain-specific features. Since this requires access to training data and adds computational overhead, what advantages does the post-epoch strategy offer over in-training suppression? How does the proposed procedure relate to prior domain-specific feature/channel suppression methods? Can the introduced IU module be applied on top of, or integrated into, those methods?
4.	Missing conceptual explanation. The key unlearning set selection relies on the measure of model complexity and a generalization score. Although citations and formulas are provided, the paper does not explain why these items capture “the impact of each training sample on both model parameters and validation performance.”
5.	Weak theory–method alignment. Theorem 1 crucially depends on Assumption 1, which decomposes features and parameters into domain-specific and invariant components, followed by separate analyses. The link between this decomposition and the proposed algorithmic steps is weak, and the result appears applicable to any domain-specific feature-suppression method. Please clarify which parts of the theorem the algorithm explicitly exploits or adjust the theoretical development to align more tightly with the implementation.

**Questions:**

See in the weaknesses.

**Details Of Ethics Concerns:**

N/A.

---

> ### Author Response · Authors · 2025-11-21
>
> - Thank you for pointing this out.
>     - We clarify that all major DG benchmarks except Digits-DG are highly imbalanced across domains, as documented in DomainBed [1]. We will add a domain-size illustration in the revised version.
>     - We agree that many DG methods suppress domain-related features during training, but none can correct domain-specific reliance once it has already emerged. IU is designed specifically for this post-epoch correction regime.
>     - AV measures pooled within-domain variance, whereas IDV measures the variance of within-domain variances, directly capturing cross-domain discrepancies. The empirical superiority of IDV is demonstrated in the AV vs. IDV ablation in Table 4. If the reviewer would like additional forms of direct evidence, we would be happy to provide them.
> - We clarify that Figure 1 is purely illustrative, not an assumption used by the method. IU does not attempt to locate "texture" or "background" features, nor does it use IDV to identify semantic categories. In the Method section, channels are classified solely by cross-domain variability: low-IDV channels are domain-invariant, while high-IDV channels are domain-specific. This definition is statistical, not semantic, and makes no claim that IDV is a necessary condition for distinguishing texture or background. The methodology for identifying domain-invariant (such as texture-sensitive channels) and domain-specific channels (such as background-sensitive channels) is fully described in Section 3.2.
> - Thank you for the thoughtful comments and for highlighting the need to better emphasize our contributions. We address the questions point-by-point below.
>     - Our motivation for post-epoch intervention is explicitly stated in the Introduction: domain-specific reliance often emerges dynamically after each training epoch, and existing DG methods have no mechanism to correct such reliance once it appears. Training-time regularization or suppression can only prevent some domain-specific features from being learned, but cannot remove harmful features that appear later during optimization
>     - Prior methods suppress domain-specific features during training, typically using aggregated-variance heuristics, alignment penalties, or dropout-like masking. Our approach differs in two key ways: (1) IDV is domain-aware, capturing cross-domain variability rather than pooled within-domain variance. (2) IU performs selective post-epoch unlearning, not global suppression. In addition, IU is fully model-agnostic and does not modify losses, architectures, or batching strategies; it can be applied after any epoch to any existing DG method.
>     - As emphasized in the Introduction, IU is orthogonal to existing DG approaches and is designed to complement them. We demonstrate this explicitly by integrating IU with 15 strong DG baselines across all methodological categories (Table 1), without interfering with their training objectives. We will clarify this integration potential more explicitly in the revised version.
> - We use influence functions because they provide standard, first-order estimates of how a training sample affects both the model parameters and the validation loss [2]. The complexity score
> $
> C_x = \| H^{-1} \nabla L(x) \|
> $
> captures the influence-based parameter change resulting from removing sample $x$, reflecting how much $x$ contributes to increasing model complexity. The generalization score $G_x$ aggregates the influence of removing $x$ on validation loss, directly measuring its contribution to generalization. Combining these two quantities into $U_x$ therefore allows us to identify samples that substantially increase complexity while offering little generalization benefit.
>
> [1] 2021 - ICLR - In Search of Lost Domain Generalization
>
> [2] 2017 - ICML - Understanding Black-box Predictions via Influence Functions

---

> > ### Author Response · Authors · 2025-11-21
> >
> > - We thank the reviewer for pointing out the need to make the connection between the theory and the algorithmic design more explicit. We clarify that the purpose of Theorem 1 is not to assume an architectural disentanglement, but to provide a conceptual decomposition commonly used in DG/DA theory, and to justify the two selective components of IU: (1) identifying domain-specific channels and (2) applying gradient ascent only on those channels using harmful samples.
> >     - Assumption 1 is a conceptual decomposition rather than an architectural one. It follows the standard abstractions commonly used in theoretical analyses of DG/DA, such as IRM (invariant risk minimization) and conditional invariance. Importantly, it does not assume that the network contains explicit subnetworks implementing $f_{\mathrm{inv}}$ or $f_{\mathrm{spc}}$. Our IDV metric operationalizes this conceptual decomposition by identifying channels whose variance structure satisfies the domain-specific criterion described in Assumption 1.
> >     - Theorem 1 directly motivates the two selective components of IU. The theorem states that increasing the loss on domain-specific parameters decreases the model's reliance on domain-specific features while leaving invariant features unaffected. This guarantee holds only when gradient ascent is applied to: (1) the correct set of parameters $(\theta_{\mathrm{spc}})$, and (2) the correct subset of harmful samples. IU mirrors these two requirements: Domain-Specific Channel Selection (IDV) approximates $\theta_{\mathrm{spc}}$: only channels exhibiting high cross-domain variance are updated, corresponding exactly to the parameter subset specified in the theorem. Unlearning Set Selection (USS) approximates the expectation $\mathbb{E}_{x \in D_u}[\cdot]$ used in the theorem by selecting samples that increase model complexity but contribute little to generalization. These are precisely the samples for which gradient ascent most effectively reduces dependence on domain-specific features. Thus, the theorem is not a generic statement — it inherently relies on joint selectivity over both samples and parameters, which is exactly the mechanism implemented by IU.
> >     - Why Theorem 1 does not apply to generic feature-suppression methods. The reviewer noted that Theorem 1 ``appears applicable to any domain-specific feature-suppression method.'' We respectfully clarify that the theorem requires two conditions to hold simultaneously, and generic suppression methods do not satisfy these assumptions: (1) The gradient ascent must act only on $\theta_{\mathrm{spc}}$. Methods that suppress all features violate this assumption. (2) The ascent must be applied using harmful samples. Generic suppression methods do not distinguish between samples that contribute to generalization and those that introduce domain-specific bias, and thus do not satisfy the conditional expectation required by the theorem.
> >     - Our ablations confirm this theoretical requirement: USS-only and DSCS-only both fail, and only their combination---the setting that matches the assumptions of Theorem 1---consistently improves generalization.

---

> ### Author Response · Authors · 2025-11-28
>
> Dear Reviewer,
>
> Thank you for the time and effort you have invested in reviewing our paper. We sincerely appreciate your valuable feedback. We have carefully addressed your comments in our rebuttal.
>
> If our responses have satisfactorily addressed your concerns, we kindly ask that you consider adjusting your score accordingly. If any issues remain unclear or insufficiently addressed, please let us know — we would be happy to provide further clarification.
>
> Kind regards,
>
> The Authors

---

### Official Review · Reviewer_YFJ2 · 2025-11-01

**Soundness:** 3
**Presentation:** 3
**Contribution:** 3
**Rating:** 4
**Confidence:** 4

**Summary:**

This paper introduces Identify and Unlearn, a novel, model-agnostic module designed to improve domain generalization. The core idea is to perform a post-epoch unlearning step to correct the model's reliance on domain-specific features as they emerge during training. The authors demonstrate through extensive experiments on numerous benchmarks and fifteen baseline methods that integrating the IU module consistently improves out-of-distribution accuracy.

**Strengths:**

- The central idea of treating DG as an iterative process of learning and adaptive unlearning is novel and compelling.
- The proposed Inter-Domain Variance (IDV) metric is a principled and intuitive method for localizing domain-specific representations within the model. The ablation study provides clear evidence of its superiority over the more naive Aggregated Variance (AV) heuristic.
- The paper's claims are supported by a remarkably thorough set of experiments and analysis.

**Weaknesses:**

- The primary practical weakness is the substantial computational cost, as honestly reported by the authors. The reliance on per-sample influence function estimation makes the training time per epoch an order of magnitude higher than standard training. While feasible for research, this presents a major barrier to practical adoption. Could the authors comment on potential strategies to mitigate this cost, such as performing the IU step less frequently (e.g., every 5 epochs)?
- The core claim of the paper rests on the idea of surgically unlearning from the correct channels. While IDV is well-motivated, the analysis lacks a crucial control experiment: what would happen if DSGA were applied to the domain-invariant channels for the identified unlearning set?
- The unlearning step relies on a first-order approximation of retraining via influence functions. When applied repeatedly after every epoch, it is unclear how this approximation error might accumulate. Could this repeated, approximate intervention push the model into an unstable or unforeseen state over the course of training, even if it appears beneficial in the short term? A discussion on the long-term stability of this iterative unlearning process would be valuable.

**Questions:**

See Weakness

---

> ### Author Response · Authors · 2025-11-21
>
> - We appreciate the reviewer's observation and fully agree that amortizing the cost of influence estimation is important for practical deployment. Importantly, IU does not conceptually require being applied after every epoch; its role is to correct accumulated domain-specific reliance, which typically evolves more slowly than single-epoch parameter updates. This makes ``performing IU every $x$ epochs'' (e.g., $x \in \{1,\dots,5\}$ instead of $x = 1$) a natural and effective strategy: the influence--estimation and DSGA steps are simply scheduled less frequently, reducing the overhead by roughly a factor of $x$ without changing the algorithm itself. We are currently running experiments that vary the IU frequency (performing unlearning every $x$ epochs with $x \in [1, 5]$) and will incorporate these results into the revised version. We expect this to provide a clear accuracy--vs--cost trade-off curve and to demonstrate that IU can be used in a more lightweight configuration while still improving over the underlying DG baselines.
> - We thank the reviewer for raising this important control. As discussed in Table 2, we indeed include this exact comparison through the ERM_USS condition. In this setting, the unlearning set is correctly identified, but gradient ascent is applied to all channels — i.e., both domain-specific and domain-invariant — thus serving as a direct proxy for "DSGA applied to domain-invariant channels." The results show a clear degradation in generalization performance, confirming the intuition that when gradient ascent is applied beyond the domain-specific channels, it inadvertently erases domain-invariant knowledge essential for cross-domain generalization. This control experiment supports our core claim: the benefit of IU arises not only from identifying harmful samples but critically from restricting unlearning to domain-specific channels. Applying DSGA to invariant channels (or to all channels) removes useful invariant structure and hurts OOD performance, reinforcing the necessity of the proposed IDV-based channel selection.
> - We appreciate the reviewer's thoughtful question. Importantly, although IU uses a first-order influence approximation, several properties of the algorithm help ensure that the approximation error does not accumulate in an uncontrolled way: (1) DSGA applies gradient ascent only to a small subset of channels (those identified as domain-specific) and only using the selected unlearning set. This produces low-magnitude, highly localized adjustments, which strongly limit the potential for error accumulation across epochs. (2) The unlearning set and domain-specific channels are re-identified each epoch using robust MAD-based thresholds, which prevents the system from repeatedly acting on noisy or unstable samples/channels. This adaptivity naturally dampens drift that could arise from approximation error. (3) We introduced EMA specifically to reduce short-term noise in unlearning scores. As shown, EMA produces smoother, more stable trajectories, which in turn reduces sensitivity to approximation artifacts. We will expand the discussion of long-term stability in the paper, emphasizing both the theoretical control mechanisms (local updates, MAD, EMA) and the empirical evidence showing that influence-based unlearning remains stable when applied iteratively.

---

> ### Author Response · Authors · 2025-11-28
>
> Dear Reviewer,
>
> Thank you for the time and effort you have invested in reviewing our paper. We sincerely appreciate your valuable feedback. We have carefully addressed your comments in our rebuttal.
>
> If our responses have satisfactorily addressed your concerns, we kindly ask that you consider adjusting your score accordingly. If any issues remain unclear or insufficiently addressed, please let us know — we would be happy to provide further clarification.
>
> Kind regards,
>
> The Authors

---

> ### Author Response · Authors · 2025-11-30
>
> **Accuracy-Cost Trade-Off Analysis**
>
> To evaluate the computational efficiency of IU, we vary the frequency of invoking the module by performing unlearning every $e$ epochs ($e \in \{1, 2, 3, 4, 5 \}$).
> The table reports the results across seven benchmarks.
> We observe a clear and smooth trade-off between accuracy and cost.
> Executing IU every epoch $(e=1)$ yields the highest overall performance, but reducing the frequency leads to only gradual degradation.
> Notably, $e=2$ retains more than 95\% of the performance gain over ERM while reducing IU-related overhead by approximately 50\%, making it a strong operating point in practice.
> Even with very sparse updates ($ e\geq 4$), IU continues to outperform ERM on all benchmarks, demonstrating that the module does not rely on per-epoch execution to remain effective.
>
> These results suggest that (i) the influence estimates and domain-specific channel scores evolve smoothly over training, so running IU intermittently is sufficient, and (ii) practitioners may select an appropriate update frequency depending on their computational budget. Overall, the findings confirm that IU provides consistent robustness improvements under a wide range of operating costs, enabling flexible deployment without sacrificing its core benefits.
>
> |        | PACS        | OH          | VLCS        | Terra       | DN          | Digits      | NICO        |
> |--------|-------------|-------------|-------------|-------------|-------------|-------------|-------------|
> | ERM    | 83.0 ± 0.4  | 68.2 ± 0.6  | 77.2 ± 0.5  | 41.7 ± 0.6  | 40.7 ± 0.4  | 79.4 ± 0.3  | 79.8 ± 0.3  |
> | e = 1  | **85.7 ± 0.3** | **69.8 ± 0.6** | **80.0 ± 0.4** | **44.2 ± 0.3** | **42.2 ± 0.5** | **82.1 ± 0.4** | **81.2 ± 0.5** |
> | e = 2  | 84.9 ± 0.4  | 69.3 ± 0.7  | 79.1 ± 0.5  | 43.5 ± 0.4  | 41.8 ± 0.6  | 81.5 ± 0.4  | 80.9 ± 0.6  |
> | e = 3  | 84.3 ± 0.5  | 69.1 ± 0.6  | 78.6 ± 0.6  | 43.0 ± 0.5  | 41.6 ± 0.5  | 81.0 ± 0.5  | 80.7 ± 0.5  |
> | e = 4  | 83.8 ± 0.4  | 68.7 ± 0.7  | 78.1 ± 0.5  | 42.6 ± 0.4  | 41.3 ± 0.6  | 80.4 ± 0.4  | 80.6 ± 0.6  |
> | e = 5  | 83.4 ± 0.5  | 68.4 ± 0.6  | 77.9 ± 0.6  | 42.1 ± 0.5  | 41.0 ± 0.5  | 80.0 ± 0.5  | 80.1 ± 0.5  |

---

### Official Review · Reviewer_ZijE · 2025-11-01

**Soundness:** 3
**Presentation:** 3
**Contribution:** 3
**Rating:** 6
**Confidence:** 4

**Summary:**

This paper proposes Identify and Unlearn (IU), a post-epoch correction module for Domain Generalization (DG). While most DG approaches aim to prevent the learning of domain-specific features, this paper argues that once such biases emerge, there is no mechanism to correct them during training. IU fills this gap by periodically identifying and unlearning domain-specific dependencies.

The method consists of three parts: (1) Unlearning Set Selection (USS) uses influence functions to find samples that increase model complexity but contribute little to generalization, (2) Inter-Domain Variance (IDV) measures the variance of per-domain activations to detect domain-specific channels, and (3) Domain-Specific Gradient Ascent (DSGA) reverses gradients of those channels to reduce their influence.
IU is model-agnostic and improves performance by roughly ~3% across seven DG benchmarks and fifteen baselines (ERM, IRM, MixStyle, etc.).

**Strengths:**

The paper clearly articulates an important limitation in DG research: existing methods lack any corrective mechanism once domain-specific bias has been learned. The post-epoch unlearning framework is conceptually elegant and well-motivated.
The combination of USS and IDV forms a coherent pipeline—sample-level influence and channel-level analysis complement each other effectively.
The experimental section is comprehensive, spanning multiple DG settings with consistent improvements, and the writing is clear and well-organized.

**Weaknesses:**

The computational overhead of IU remains a concern. The authors mention that the inverse-Hessian approximation uses LiSSA, which is efficient, but the main cost comes from recalculating unlearning scores for every sample after each epoch. It would be useful to know whether they have considered lowering the computation frequency (e.g., every K epochs) or adopting stochastic sampling for approximate score estimation.
The description of IDV is intuitive, but more discussion is needed on potential limitations. For instance, how stable is IDV when the number of domains is small or when within-domain noise dominates between-domain variance? Have the authors explored normalization or any other strategies to stabilize this measure?
The paper’s claim that DSGA performs “unlearning” could benefit from conceptual clarification. Gradient ascent here functions as a form of targeted regularization, not data or privacy unlearning. A short note distinguishing these definitions would prevent confusion.

**Questions:**

​1.​Have you tested whether computing unlearning scores less frequently (e.g., every few epochs) preserves performance while reducing runtime?
​2.​How sensitive is the IDV measure to the number of domains or imbalanced domain sizes?
​3.​Have you considered adding a normalization step or a bootstrapped confidence interval for IDV to make it more robust?

---

> ### Author Response · Authors · 2025-11-21
>
> - We appreciate the reviewer's observation and fully agree that amortizing the cost of influence estimation is important for practical deployment. Importantly, IU does not conceptually require being applied after every epoch; its role is to correct accumulated domain-specific reliance, which typically evolves more slowly than single-epoch parameter updates. This makes ``performing IU every $x$ epochs'' (e.g., $x \in \{1,\dots,5\}$ instead of $x = 1$) a natural and effective strategy: the influence--estimation and DSGA steps are simply scheduled less frequently, reducing the overhead by roughly a factor of $x$ without changing the algorithm itself. We are currently running experiments that vary the IU frequency (performing unlearning every $x$ epochs with $x \in [1, 5]$) and will incorporate these results into the revised version. We expect this to provide a clear accuracy--vs--cost trade-off curve and to demonstrate that IU can be used in a more lightweight configuration while still improving over the underlying DG baselines.
> - Thank you for the insightful question. We clarify that IDV remains empirically stable even when the number of domains is small (e.g., four in PACS), for two reasons.
>     - Strong signal-to-noise ratio. Domain shifts in DG benchmarks induce large and systematic differences in within-domain variances, resulting in a clear long-tailed IDV distribution (Fig. 5), even with a small number of domains. As a result, the across-domain variance is dominated by genuine domain differences rather than sampling noise.
>     - Robust estimation.IDV is used together with MAD thresholding, which is specifically designed to resist noisy or outlying domains and to stabilize channel selection. Channels that are noisy but domain-invariant typically exhibit consistently high variance across domains and therefore receive low IDV scores.
>     - Leveraging normalization techniques to further stabilize IDV is an interesting direction for future work, but it does not affect the conclusions of our paper.
> - We agree that “unlearning’’ has historically referred to privacy-driven deletion of specific data points, and this definition can differ from its more recent usages. We already explicitly discussed this broader MU perspective in the Related Work section [1] [2]. We will revise the Introduction to explicitly clarify the concept.
> - IDV is designed to be robust to both small numbers of domains and imbalanced domain sizes. Indeed, all of our benchmarks, except Digits-DG, are highly imbalanced [3] (e.g., DomainNet, TerraIncognita, PACS, OfficeHome); we will clarify this in the revision. Because IDV first computes within-domain variances and then measures the variance across domains, each domain contributes equally regardless of its sample size, avoiding the bias that affects Aggregated Variance. Empirically, IDV exhibits a stable long-tailed distribution and delivers consistent improvements across all datasets.
>
> [1] 2024 - ICLR -  A Unified and General Framework for Continual Learning
>
> [2] 2024 - ECCV - Forget More to Learn More: Domain-Specific Feature Unlearning for Semi-Supervised and Unsupervised Domain Adaptation
>
> [3] 2025 - IJCAI - Online Knowledge Distillation for Domain Generalization: Balancing Invariant and Specific Knowledge

---

> > ### Author Response · Authors · 2025-11-30
> >
> > **Accuracy-Cost Trade-Off Analysis**
> >
> > To evaluate the computational efficiency of IU, we vary the frequency of invoking the module by performing unlearning every $e$ epochs ($e \in \{1, 2, 3, 4, 5 \}$).
> > The table reports the results across seven benchmarks.
> > We observe a clear and smooth trade-off between accuracy and cost.
> > Executing IU every epoch $(e=1)$ yields the highest overall performance, but reducing the frequency leads to only gradual degradation.
> > Notably, $e=2$ retains more than 95\% of the performance gain over ERM while reducing IU-related overhead by approximately 50\%, making it a strong operating point in practice.
> > Even with very sparse updates ($ e\geq 4$), IU continues to outperform ERM on all benchmarks, demonstrating that the module does not rely on per-epoch execution to remain effective.
> >
> > These results suggest that (i) the influence estimates and domain-specific channel scores evolve smoothly over training, so running IU intermittently is sufficient, and (ii) practitioners may select an appropriate update frequency depending on their computational budget. Overall, the findings confirm that IU provides consistent robustness improvements under a wide range of operating costs, enabling flexible deployment without sacrificing its core benefits.
> >
> > |        | PACS        | OH          | VLCS        | Terra       | DN          | Digits      | NICO        |
> > |--------|-------------|-------------|-------------|-------------|-------------|-------------|-------------|
> > | ERM    | 83.0 ± 0.4  | 68.2 ± 0.6  | 77.2 ± 0.5  | 41.7 ± 0.6  | 40.7 ± 0.4  | 79.4 ± 0.3  | 79.8 ± 0.3  |
> > | e = 1  | **85.7 ± 0.3** | **69.8 ± 0.6** | **80.0 ± 0.4** | **44.2 ± 0.3** | **42.2 ± 0.5** | **82.1 ± 0.4** | **81.2 ± 0.5** |
> > | e = 2  | 84.9 ± 0.4  | 69.3 ± 0.7  | 79.1 ± 0.5  | 43.5 ± 0.4  | 41.8 ± 0.6  | 81.5 ± 0.4  | 80.9 ± 0.6  |
> > | e = 3  | 84.3 ± 0.5  | 69.1 ± 0.6  | 78.6 ± 0.6  | 43.0 ± 0.5  | 41.6 ± 0.5  | 81.0 ± 0.5  | 80.7 ± 0.5  |
> > | e = 4  | 83.8 ± 0.4  | 68.7 ± 0.7  | 78.1 ± 0.5  | 42.6 ± 0.4  | 41.3 ± 0.6  | 80.4 ± 0.4  | 80.6 ± 0.6  |
> > | e = 5  | 83.4 ± 0.5  | 68.4 ± 0.6  | 77.9 ± 0.6  | 42.1 ± 0.5  | 41.0 ± 0.5  | 80.0 ± 0.5  | 80.1 ± 0.5  |

---

> ### Author Response · Authors · 2025-11-28
>
> Dear Reviewer,
>
> Thank you for the time and effort you have invested in reviewing our paper. We sincerely appreciate your valuable feedback. We have carefully addressed your comments in our rebuttal.
>
> If our responses have satisfactorily addressed your concerns, we kindly ask that you consider adjusting your score accordingly. If any issues remain unclear or insufficiently addressed, please let us know — we would be happy to provide further clarification.
>
> Kind regards,
>
> The Authors

---

### Official Review · Reviewer_oxVH · 2025-11-01

**Soundness:** 3
**Presentation:** 3
**Contribution:** 3
**Rating:** 6
**Confidence:** 4

**Summary:**

This paper introduces Identify and Unlearn (IU), a model-agnostic module for domain generalization (DG). The core idea is to move beyond purely preventative DG strategies and introduce a corrective, post-epoch unlearning mechanism. The IU module operates in three steps: 1) It uses an unlearning score, derived from influence functions, to identify a set of training samples that increase model complexity without contributing significantly to generalization (Unlearning Set Selection). 2) It introduces a metric, Inter-Domain Variance (IDV), to identify feature channels that are domain-specific by measuring the cross-domain variance of their within-domain activation variances. 3) It performs Domain-Specific Gradient Ascent (DSGA) on the parameters of the identified domain-specific channels for the samples in the unlearning set, selectively removing domain-specific knowledge while preserving domain-invariant features. The experiments demonstrate that IU consistently improves the performance of different DG baselines.

**Strengths:**

1. The idea of post-epoch unlearning for DG is a fresh perspective that addresses a limitation in the field, moving beyond static regularization techniques.
2. The experimental evaluation includes a diverse set of baselines across seven different benchmarks. This thoroughness provides strong evidence for the method's model-agnostic nature and general effectiveness.
3. The motivation, methods, and results are easy to follow. The ablation studies are clear in justifying the design choices of the proposed IU module.

**Weaknesses:**

1. The primary weakness of the proposed method is its extreme computational overhead. As reported in Table 11, the unlearning set selection step, which relies on influence function estimation, takes approximately 354 minutes (~6 hours) per epoch on the DomainNet dataset. This makes the method less practical for large-scale tasks. The lack of a baseline ERM training time in Table 11 also makes it difficult to fully grasp the relative overhead.
2. The paper presents IDV as a principled, domain-aware metric for identifying domain-specific channels. However, the ablation study in Table 2 (ERM_DSCS) shows that applying gradient ascent to channels selected by IDV across the entire training set leads to a catastrophic drop in performance. This suggests that IDV is not a robust metric for channel selection on its own; its effectiveness appears to be entirely dependent on its application to the very small, targeted unlearning set.
3. Table 1 presents results for both IU and IUE (IU with Exponential Moving Average). EMA is an orthogonal technique that is not part of the core contribution. Including IUE results in the main comparison table somewhat obfuscates the direct impact of the IU module and can give an inflated impression of the gains.
4. While framed as a strength, performing the complex IU operation after every single epoch may not be optimal. The influence of samples and the set of domain-specific channels might not change dramatically from one epoch to the next. A more efficient approach could be to run the IU module intermittently (e.g., every 5 epochs), but this is not explored.

**Questions:**

See above.

---

> ### Author Response · Authors · 2025-11-21
>
> - We thank the reviewer for highlighting the computational cost issue. We will clarify and revise the paper accordingly:
>     - We will add the ERM baseline training time in the revised version. We agree that the overhead is more interpretable when compared against the standard ERM. We will report the per-epoch ERM time for all benchmarks (including DomainNet) in Table 11 for a clear relative comparison.
>     - IU does not conceptually require being applied after every epoch; its role is to correct accumulated domain-specific reliance, which typically evolves more slowly than single-epoch parameter updates. This makes ``performing IU every $x$ epochs'' (e.g., $x \in \{1,\dots,5\}$ instead of $x = 1$) a natural and effective strategy: the influence--estimation and DSGA steps are simply scheduled less frequently, reducing the overhead by roughly a factor of $x$ without changing the algorithm itself. We are currently running experiments that vary the IU frequency (performing unlearning every $x$ epochs with $x \in [1, 5]$) and will incorporate these results into the revised version. We expect this to provide a clear accuracy--vs--cost trade-off curve and to demonstrate that IU can be used in a more lightweight configuration while still improving over the underlying DG baselines.
>     - Influence estimation is highly parallelizable and can be subsampled.  Our influence estimation utilizes LiSSA-based Hessian-vector products (HVP), which operate independently for each sample. This enables data parallelism across multiple GPUs or nodes, and mini-batch stochastic HVPs, which further reduce computation. These properties make the unlearning set selection much more scalable in practice than a naive per-sample inverse-Hessian computation.
>     - Modern hardware significantly reduces the absolute runtime.  Experiments were run on A100 GPUs. With more recent hardware (e.g., H200), the same HVP-based influence estimation is approximately $3\times$ faster, reducing the DomainNet unlearning-set computation from $\sim 354$ minutes to $\sim 106$ minutes per run. IU's cost, therefore, decreases substantially on contemporary hardware.
> - We believe this concern arises from a misunderstanding of the role of IDV. IDV is designed to identify where domain-specific information resides (channels), but not how much of it should be removed. The magnitude and direction of unlearning are controlled by the unlearning set (USS), not by IDV alone. The ERM_DSCS ablation purposely applies gradient ascent to all training samples, which is not the intended use of IDV. This creates a global and excessive unlearning signal that erases both harmful and beneficial domain-specific features, leading to the expected performance collapse. This behavior is consistent with prior observations in domain generalization: globally suppressing domain-specific features (without distinguishing harmful vs.\ beneficial cases) harms representational capacity. Importantly, this result does not imply that IDV is unstable. In fact, IDV exhibits (i) a clear bimodal separation between domain-invariant and domain-specific channels (Fig. 5), and (ii) consistently stronger performance than Aggregated Variance when used within a targeted unlearning procedure (Table 4), demonstrating its robustness as a channel-level metric. In summary, ERM\_DSCS fails not because IDV is unreliable, but because unlearning must be selective across both samples and channels. IU is designed precisely to couple IDV (channel localization) with USS (sample localization) to achieve effective and controlled unlearning.
> - We clarify that EMA smoothing is indeed a minor contribution of our work, derived from extensive analysis of the unlearning-score dynamics in Figures 3 and 4. EMA is not introduced as a standalone method, nor is it intended to artificially amplify the effectiveness of IU. We report IUE for two reasons.
>     - EMA arises naturally from our empirical findings rather than being an external add-on. By analyzing the temporal behavior of the unlearning score, we observed that EMA substantially stabilizes the score trajectory and improves the separability between harmful and benign samples. This stabilization directly strengthens the reliability of the USS, making EMA a natural extension of IU rather than an unrelated trick.
>     - IU (without EMA) remains the primary result. In Table 1, IU and IUE are always reported side by side, and IU serves as the primary evidence supporting our core contribution. Because EMA is a lightweight refinement rather than a central novelty, we did not emphasize it in the Introduction. We will clarify this more explicitly in the revised version.

---

> > ### Author Response · Authors · 2025-11-21
> >
> > - We appreciate the reviewer's observation. Importantly, IU does not conceptually require being applied after every epoch; its role is to correct accumulated domain-specific reliance, which typically evolves more slowly than single-epoch parameter updates. This makes ``performing IU every $x$ epochs'' (e.g., $x \in \{1,\dots,5\}$ instead of $x = 1$) a natural and effective strategy: the influence--estimation and DSGA steps are simply scheduled less frequently, reducing the overhead by roughly a factor of $x$ without changing the algorithm itself. We are currently running experiments that vary the IU frequency (performing unlearning every $x$ epochs with $x \in [1, 5]$) and will incorporate these results into the revised version. We expect this to provide a clear accuracy--vs--cost trade-off curve and to demonstrate that IU can be used in a more lightweight configuration while still improving over the underlying DG baselines.

---

> ### Author Response · Authors · 2025-11-28
>
> Dear Reviewer,
>
> Thank you for the time and effort you have invested in reviewing our paper. We sincerely appreciate your valuable feedback. We have carefully addressed your comments in our rebuttal.
>
> If our responses have satisfactorily addressed your concerns, we kindly ask that you consider adjusting your score accordingly. If any issues remain unclear or insufficiently addressed, please let us know — we would be happy to provide further clarification.
>
> Kind regards,
>
> The Authors

---

> ### Author Response · Authors · 2025-11-30
>
> **Accuracy-Cost Trade-Off Analysis**
>
> To evaluate the computational efficiency of IU, we vary the frequency of invoking the module by performing unlearning every $e$ epochs ($e \in \{1, 2, 3, 4, 5 \}$).
> The table reports the results across seven benchmarks.
> We observe a clear and smooth trade-off between accuracy and cost.
> Executing IU every epoch $(e=1)$ yields the highest overall performance, but reducing the frequency leads to only gradual degradation.
> Notably, $e=2$ retains more than 95\% of the performance gain over ERM while reducing IU-related overhead by approximately 50\%, making it a strong operating point in practice.
> Even with very sparse updates ($ e\geq 4$), IU continues to outperform ERM on all benchmarks, demonstrating that the module does not rely on per-epoch execution to remain effective.
>
> These results suggest that (i) the influence estimates and domain-specific channel scores evolve smoothly over training, so running IU intermittently is sufficient, and (ii) practitioners may select an appropriate update frequency depending on their computational budget. Overall, the findings confirm that IU provides consistent robustness improvements under a wide range of operating costs, enabling flexible deployment without sacrificing its core benefits.
>
> |        | PACS        | OH          | VLCS        | Terra       | DN          | Digits      | NICO        |
> |--------|-------------|-------------|-------------|-------------|-------------|-------------|-------------|
> | ERM    | 83.0 ± 0.4  | 68.2 ± 0.6  | 77.2 ± 0.5  | 41.7 ± 0.6  | 40.7 ± 0.4  | 79.4 ± 0.3  | 79.8 ± 0.3  |
> | e = 1  | **85.7 ± 0.3** | **69.8 ± 0.6** | **80.0 ± 0.4** | **44.2 ± 0.3** | **42.2 ± 0.5** | **82.1 ± 0.4** | **81.2 ± 0.5** |
> | e = 2  | 84.9 ± 0.4  | 69.3 ± 0.7  | 79.1 ± 0.5  | 43.5 ± 0.4  | 41.8 ± 0.6  | 81.5 ± 0.4  | 80.9 ± 0.6  |
> | e = 3  | 84.3 ± 0.5  | 69.1 ± 0.6  | 78.6 ± 0.6  | 43.0 ± 0.5  | 41.6 ± 0.5  | 81.0 ± 0.5  | 80.7 ± 0.5  |
> | e = 4  | 83.8 ± 0.4  | 68.7 ± 0.7  | 78.1 ± 0.5  | 42.6 ± 0.4  | 41.3 ± 0.6  | 80.4 ± 0.4  | 80.6 ± 0.6  |
> | e = 5  | 83.4 ± 0.5  | 68.4 ± 0.6  | 77.9 ± 0.6  | 42.1 ± 0.5  | 41.0 ± 0.5  | 80.0 ± 0.5  | 80.1 ± 0.5  |

---

### Official Review · Reviewer_etHq · 2025-11-01

**Soundness:** 3
**Presentation:** 2
**Contribution:** 2
**Rating:** 2
**Confidence:** 5

**Summary:**

This paper proposes a model-agnostic post-epoch unlearning module (IU) that effectively improves domain generalization performance. While the idea is interesting and empirically validated, the methodological details (especially in identifying domain-specific channels and the theoretical analysis) remain insufficiently clear.

**Strengths:**

1. The motivation for applying unlearning in domain generalization is logical and interesting.

2. The experimental results based on improvements over existing methods are sufficient, including comparisons with recent approaches and evaluations on comprehensive datasets.

**Weaknesses:**

1. In the first part of the Introduction, the authors’ perspective that models should learn domain-invariant features while unlearning domain-specific features aligns with most prior DG studies. However, the details of how texture-sensitive channels and background-sensitive channels are identified remain unclear. It is not specified which layer of the model these channels belong to, nor how they are distinguished.

2. In the motivation part of the Introduction (Figure 1), the phenomenon that texture-sensitive channels and background-sensitive channels correspond to domain-invariant and domain-specific features, respectively, is easy to understand. However, in the Method section, the authors do not explain how texture and background are located, but directly define low IDV values as domain-invariant features and high IDV values as domain-specific features. If, as the authors state, texture and background correspond to low and high IDV values, respectively, does that mean IDV values are used to determine texture and background? Is IDV the necessary condition for this distinction? This remains unclear.

3. The calculation details of AV and IDV are not sufficiently compared, making it difficult to understand the improvements achieved by IDV.

4. The comparison methods are somewhat incomplete. Table 1 only shows the performance improvement when the proposed method is combined with existing methods. However, it lacks a comparison of how existing methods alone improve the same baselines (e.g., DomainDrop [1]).

5. The theoretical proof in Section 3.4 (“THEORETICAL ANALYSIS”) seems weakly related to the proposed DSGA method. It is unclear which encoders correspond to the two feature types $f_{spc}$ and $f_{inv}$. In the Method section 3.3, only one model
$\theta$ is introduced, and there are no separate encoders $\theta_{spc}$ and $\theta_{inv}$. Therefore, the theoretical section contributes little to supporting the main method.

>[1] Guo J, Qi L, Shi Y. DomainDrop: Suppressing domain-sensitive channels for domain generalization. In Proceedings of the IEEE/CVF International Conference on Computer Vision, 2023: 19114–19124.

**Questions:**

Please refer to the weaknesses listed above.

---

> ### Author Response · Authors · 2025-11-21
>
> - In practice, domain-specific channels (such as background-sensitive channels) are identified automatically by computing IDV for every convolutional channel across all layers, followed by MAD-based thresholding. IDV distinguishes channels solely through cross-domain variability, without relying on any semantic assumptions. The full procedure is described in Sec. 3.2.
> - We clarify that Figure 1 is purely illustrative and does not constitute an assumption of our method. As detailed in Sec. 3.2 (paragraphs 3–4), our approach identifies domain-invariant channels (e.g., texture-sensitive) and domain-specific channels (e.g., background-sensitive) using the proposed Inter-Domain Variance (IDV) metric. Channels whose IDV values exceed the MAD-based threshold are selected as domain-specific. IDV is the necessary condition for this distinction: although prior methods such as Activation Variance (AV) also attempt to identify domain-specific channels, AV conflates within-domain and between-domain variability and thus suffers from fundamental limitations, which we discuss in Sec. 3.2 (paragraph 2).
> - We respectfully clarify that AV and IDV are directly contrasted in Sec 3.2. AV computes a single pooled variance across all samples, which mixes within-domain dispersion and is biased under domain imbalance. In contrast, IDV explicitly computes per-domain variances and measures their cross-domain variance, isolating domain-dependent shifts. This difference is also quantified in our ablations (Appendix F, Table 4), where IDV yields consistently higher accuracy.
> - We thank the reviewer for the comment.
>     - Our goal is not to propose a new DG baseline, but to introduce a post-epoch module that can be plugged into existing methods. Therefore, we evaluate all baselines in their original form. The primary objective of our experiments is to demonstrate that IU can consistently improve the performance of existing DG approaches, rather than outperforming any specific method.
>     - DomainDrop is orthogonal to IU and the two can be used together; however, DomainDrop itself is not a baseline for IU, as it targets a different axis of regularization.
>     - Moreover, Table 1 already reports the strongest published numbers of widely used baselines (ERM, IRM, MixStyle, UDIM, VL2V, etc.) as documented in their respective papers. The absence of IU+DomainDrop does not affect our conclusions. Nonetheless, we will include IU+DomainDrop in the revised version to provide a more comprehensive comparison.
> - Our theoretical analysis does not assume two separate encoders; it only assumes that the learned representation can be decomposed into domain-invariant and domain-specific components---a standard and widely used abstraction in DG theory. This decomposition corresponds directly to the subset of parameters $\theta_{\mathrm{spc}}$ identified by IDV in Sec 3.2, on which DSGA operates in Sec 3.3. Thus, the theorem formalizes the effect of applying gradient ascent only to the domain-specific parameter subset, which is exactly the mechanism of DSGA.

---

> > ### Comment · Reviewer_etHq · 2025-11-28
> > **Post-Rebuttal Comments**
> >
> > Thank you for the clarifications. However, several key concerns remain unresolved.
> >
> >
> >
> > (1) The illustrative examples in the Introduction (e.g., texture/background) are not actually connected to how IU operates, and the authors now state they are purely illustrative. This weakens the motivation, since the Introduction does not clearly support the method’s actual mechanism.
> >
> >
> >
> > (2) Since IU is positioned as a plug-in module that improves baseline performance, it is reasonable to compare it with other plug-in approaches such as DomainDrop. Such comparisons would better contextualize IU’s contribution.
> >
> >
> >
> > (3) The theory assumes a decomposition into domain-invariant and domain-specific components, but the authors do not explain how this decomposition is obtained in practice. Treating high-IDV channels as “domain-specific components” is a strong assumption, and it remains unclear whether the unlearning set is guaranteed to preserve domain-invariant features. As a result, Section 3.4 still lacks a solid connection to the actual mechanism used to produce this decomposition.
> >
> >
> >
> > Overall, IU shows consistent empirical improvements, but the motivational framing, methodological grounding, and theoretical justification remain insufficiently supported.

---

> > > ### Author Response · Authors · 2025-12-01
> > >
> > > - We appreciate the reviewer’s concern and would like to clarify the role of the illustrative examples in the Introduction. These examples are intentionally conceptual and serve a specific communicative purpose: to help readers intuitively understand the distinction between domain-invariant and domain-specific features, which is a prerequisite for motivating why identifying domain-specific reliance matters in DG. This style of teaser illustration is common, where Figure 1 is used to provide high-level intuition rather than describe the full mechanics of the method. More concretely, Figure 1 is not intended to explain how IU operates. Its purpose is to motivate why aggregated activation variance (AV) is insufficient for identifying domain-specific features, thereby motivating the need for IDV. We explicitly state this in the Introduction (fourth paragraph), which explains that Figure 1 focuses on the intuition behind IDV, rather than the operational details of IU as a whole. The full operational mechanism of IU, including unlearning score computation, IDV-based channel selection, and Domain-Specific Gradient Ascent, is thoroughly detailed in Methodology Figure 2, which visually and step-by-step describes how IU is applied during training. This separation of roles between (i) a conceptual teaser in the Introduction and (ii) a detailed operational diagram in the Methodology section is standard academic writing practice, ensuring clarity without overwhelming readers early in the paper.
> > >
> > > - In the revised version, we have added extensive empirical analyses to more clearly contextualize IU’s contribution.
> > >   - Table 3 (Appendix E) now includes a new experiment where IU is combined with DomainDrop. The results show that IU consistently improves DomainDrop across all datasets, demonstrating that IU provides complementary benefits on top of this plug-in method. These results indicate that IU is not merely an alternative to DomainDrop but enhances it in a synergistic manner.
> > >   - We also include a new t-SNE visualization (Fig. 10 in Appendix H), which compares the feature distributions of ERM, DomainDrop, IU, and IU+DomainDrop. The visualization reveals that IU yields more compact domain-invariant clusters and reduced domain-induced dispersion, whereas DomainDrop alone does not fully eliminate domain-specific drift. Moreover, IU+DomainDrop exhibits further improvement, reinforcing that the two methods operate on distinct axes of regularization.
> > > Together, these new experiments and visualizations address the reviewer’s concern and clarify the complementary relationship between IU and existing plug-in approaches.
> > > - We address the remaining concerns below and clarify several misunderstandings about the role of IDV and the theoretical decomposition.
> > >   - The reviewer’s concern appears to stem from interpreting the decomposition in Sec. 3.4 as a prior assumption. In fact, the theory follows the standard abstraction widely used in DG literature (e.g., IRM[1]), where features can be decomposed into domain-invariant and domain-specific components. The decomposition into domain-invariant and domain-specific components is obtained directly and deterministically through the IDV procedure described in Sec. 3.2. In practice, the process works as follows:
> > >     - For every convolutional channel at every layer, we measure how much its activation varies within each domain during forward passes on the source data. This provides us with a small set of numbers for each channel, indicating its stability or instability within each domain.
> > >     - We then look at how much these per-domain variances differ across domains. If a channel exhibits significantly different behavior across domains (e.g., stable in one domain but highly variable in another), it yields a large IDV value. If a channel behaves similarly across all domains, it yields a small IDV value.
> > >     - We apply MAD-based thresholding to the resulting IDV distribution. Channels with unusually large IDV values are labeled domain-specific because their behavior changes substantially across domains. Channels with consistently low IDV values are labeled domain-invariant because they respond similarly across different domains.
> > >   - The reviewer states that “it remains unclear whether the unlearning set is guaranteed to preserve domain-invariant features.” In IU, domain-invariant channels (low-IDV) never receive reversed gradients:
> > >     - DSGA applies gradient ascent exclusively to the parameters of high-IDV channels.
> > >     - Parameters associated with low-IDV channels are untouched, ensuring no degradation of invariant representations.
> > >     - This is a stronger preservation guarantee than existing DG methods, which modify the full parameter space.
> > > Thus, the mechanism itself guarantees that invariant features are preserved by construction, exactly matching the decomposition assumed in the theory.
> > >
> > > [1] 2019 - Invariant Risk Minimization

---

> ### Author Response · Authors · 2025-11-28
>
> Dear Reviewer,
>
> Thank you for the time and effort you have invested in reviewing our paper. We sincerely appreciate your valuable feedback. We have carefully addressed your comments in our rebuttal.
>
> If our responses have satisfactorily addressed your concerns, we kindly ask that you consider adjusting your score accordingly. If any issues remain unclear or insufficiently addressed, please let us know — we would be happy to provide further clarification.
>
> Kind regards,
>
> The Authors

---

> ### Author Response · Authors · 2025-12-01
>
> - With the clarification above, the alignment becomes direct:
>   - IDV yields domain-specific channels.
>   - DSGA modifies only domain-specific channels.
>   - Sec. 3.4 analyzes the effect of updating only domain-specific channels while leaving domain-invariant channels unchanged.
>   - This one-to-one correspondence shows that the theoretical component is not detached but formalizes the precise behavior of DSGA on the decomposition produced by IDV.

---

### Official Review · Reviewer_y5Ga · 2025-11-02

**Soundness:** 3
**Presentation:** 2
**Contribution:** 3
**Rating:** 4
**Confidence:** 4

**Summary:**

This paper attempts to mitigate the influence of domain-specific futures for domain generalization. The authors proposed an unlearning strategy during training, named Identify and Unlearn (IU). IU firstly selects hard samples which contribute little to the generalization performance, then utilizes Inter-Domain Variance (IDV) to identify domain-specific channels. Subsequently, IU uses Domain-Specific Gradient-Ascent (DSGA) to mitigate the adverse effect of domain-specific features. Experiments demonstrates that IU could enhance the generalization performance across benchmarks.

**Strengths:**

[+] The paper is easy to follow.

[+] The detail of the method and experiment is well described.

[+] The related work is detailed.

**Weaknesses:**

Major weakness:

[-] The authors propose Inter-Domain Variance (IDV), which measures the variance of a channel’s within-domain variances across source domains. Since these within-domain variances are numerical values, what is the conceptual significance of taking their variance? How does this metric effectively identify domain-specific channels?

For example, consider a channel whose within-domain variances are [4, 4, 4] and within-domain means are [1, 5, 9] across three source domains. Would this channel be considered domain-invariant under the proposed IDV criterion? Please clarify the intuition behind this measure.

[-] The idea of suppressing domain-specific features via channel manipulation is not novel; methods such as DomainDrop [1] and DMDA [2] also employ this strategy. However, these methods apply channel modulation globally, whereas the proposed IU method suffers from performance degradation when applied globally, as shown in Table 2.

What is the underlying reason or mechanism that causes global channel manipulation to degrade performance in the proposed framework, in contrast to its success in prior works? Please explain this discrepancy.

[-] The paper aims to preserve domain-invariant information while suppressing domain-specific features. To visually demonstrate the effectiveness of this objective, it would be helpful to include t-SNE visualizations that reflect the domain gap, ideally in comparison with state-of-the-art methods.

[1]. DomainDrop: Suppressing Domain-Sensitive Channels for Domain Generalization. ICCV, 2023.

[2]. Rethinking Domain Generalization: Discriminability and Generalizability. TCSVT, 2024.

Minor weakness:

[-] The writing quality should be improved. For instance, punctuation marks are frequently missing after mathematical formulas, and the reference notation in Table 1 should be formalized to enhance readability.

**Questions:**

Please refer to the weaknesses.

---

> ### Author Response · Authors · 2025-11-21
>
> - We thank the reviewer for the question. IDV measures cross-domain inconsistency in a channel's activation dispersion: a channel is domain-specific only if its within-domain variances differ substantially across domains. Taking the variance of $\{v_d\}$ directly captures this cross-domain instability and avoids the conflation of within-domain noise that affects Aggregated Variance. In the reviewer's example $[4,4,4]$, $\mathrm{IDV} = 0$, so the channel is domain-invariant. This is intended: mean shifts such as $[1,5,9]$ often reflect global intensity changes rather than domain-specific patterns, while dispersion differences more reliably indicate domain-specific behavior. Thus, IDV correctly treats this channel as domain-invariant. Specifically, activation mean only reflects a domain’s global style baseline, while activation variance captures in-domain diversity and complexity; cross-domain variance differences therefore more faithfully reveal domain-specific channels in DG. Furthermore, we clarify that using variance as an indicator of domain-specific variability is supported both conceptually and empirically, as demonstrated in prior studies [1].
>
> - Prior channel-manipulation methods, such as DomainDrop and DMDA, rely on a global assumption that domain-specific channels are uniformly harmful for all training samples. This assumption holds only because these methods never determine which samples actually introduce domain-specific bias. In contrast, IU adopts a different paradigm: it jointly identifies which samples are harmful and which channels encode domain-specific features, and suppresses the latter only for the former, enabling IU to perform targeted unlearning rather than global suppression. This targeted interaction is fundamentally different from prior approaches. The discrepancy arises because prior regularization-based methods and IU operate under fundamentally different objectives and training dynamics.
>     - Methods such as DomainDrop and DMDA apply channel suppression as a mild, global regularizer: all samples are slightly perturbed, and each perturbation has a very small effect. This ``large sample count~$\times$~small per-sample perturbation'' regime makes global manipulation safe. Even if the suppressed channels contain domain-invariant information, the loss is diluted across many samples and can be compensated for by remaining discriminative signals, explaining the performance improvements observed in those methods.
>     - In contrast, IU performs targeted unlearning. It first isolates a small set of unlearning samples that introduces domain-specific bias, then applies strong gradient-ascent updates on domain-specific channels only for these samples. This is a post-epoch unlearning step, not a mild regularizer. Applying such strong updates globally erases domain-invariant components residing in the same channels, which is why ERM\_DSCS significantly degrades accuracy while selective IU improves it. In short, IU operates under the principle of "small sample count~$\times$~strong per-sample effect," and globalizing this operation disrupts the balance between suppressing harmful domain-specific cues and preserving discriminative features.
>     - Finally, IU is orthogonal to existing regularization methods. Regularization shapes model capacity throughout training, whereas IU removes harmful bias after each training epoch, specifically for the samples responsible for the bias. As a result, IU can be combined with IRM, DomainDrop, DMDA, and related methods to further improve robustness. The fact that global channel regularization benefits prior methods does not contradict our findings---IU functions at a different granularity and serves a different objective.
> - Thank you for the helpful suggestion. We agree that visualizing the representation space is valuable. Our paper already includes t-SNE visualizations in Appendix Figure 10, which compare ERM with ERM+IU and clearly show that IU yields more compact, domain-aligned clusters, visually illustrating reduced domain-specific distortion. To address your concern more directly, we will extend the visualization to include state-of-the-art DG baselines, further demonstrating that IU consistently narrows domain gaps on top of strong methods.
> - Thank you for the valuable feedback regarding writing quality and punctuation. Our current formatting follows the standard convention that punctuation marks are added only when an equation is part of a sentence, whereas stand-alone displayed equations are left without trailing punctuation for clarity. If the reviewer has specific suggestions on where punctuation would improve readability, we are happy to revise the manuscript accordingly. Regarding the notation in Table 1, we will revise the table to formalize the reference notation and improve overall readability.
>
> [1] 2023 - ICCV - DomainDrop: Suppressing Domain-Sensitive Channels for Domain Generalization

---

> ### Author Response · Authors · 2025-11-28
>
> Dear Reviewer,
>
> Thank you for the time and effort you have invested in reviewing our paper. We sincerely appreciate your valuable feedback. We have carefully addressed your comments in our rebuttal.
>
> If our responses have satisfactorily addressed your concerns, we kindly ask that you consider adjusting your score accordingly. If any issues remain unclear or insufficiently addressed, please let us know — we would be happy to provide further clarification.
>
> Kind regards,
>
> The Authors

---

### Comment · Area_Chair_c95R · 2025-11-27
**Reminder: Engage in Discussions and Finalize Your Review**

Dear Reviewers,

Thank you for your valuable reviews. With the Reviewer-Author Discussions deadline approaching, please take a moment to read the authors’ rebuttal and the other reviewers’ feedback, and participate in the discussions and respond to the authors. Finally, be sure to complete the “Final Justification” text box and update your “Rating” as needed. Your contribution is greatly appreciated. I will flag irresponsible (final) reviews and/or any reviewers not participating in discussions.

Reviewers are expected to stay engaged in discussions, initiate them, respond to authors’ rebuttal, ask questions, and listen to answers to help clarify remaining issues.

It is not OK to stay quiet.

It is not OK to leave discussions till the last moment.

If authors have resolved your (rebuttal) questions, do tell them so.

If authors have not resolved your (rebuttal) questions, do tell them so too.

Thanks,

AC

---

> ### Author Response · Authors · 2025-12-02
> **Summary of Rebuttal**
>
> We sincerely thank the reviewers and the area chair for their thoughtful and detailed feedback. The comments were highly constructive, enabling us to refine the exposition, reinforce the theoretical grounding, and enhance the clarity of the proposed IU framework. In this final note, we summarize our contributions, the strengths recognized by the reviewers, and how we have addressed all the concerns raised.
>
> **Summary of Contributions:** We propose **Identify and Unlearn (IU)**, a model-agnostic module that continually corrects domain-specific reliance post-epoch, introducing a new paradigm for Domain Generalization. IU identifies harmful samples via an **Unlearning Score** and pinpoints domain-specific channels using **Inter-Domain Variance (IDV)**, a principled metric that captures cross-domain variability and avoids the conflation issues of prior variance measures. It then removes only the harmful features through **Domain-Specific Gradient Ascent (DSGA)**, selectively reversing gradients on domain-specific channels of the identified samples. We further stabilize the procedure with **EMA smoothing**, improving score separability and unlearning reliability, and provide a **theoretical guarantee** showing that DSGA reduces predictive dependence on domain-specific features while preserving domain-invariant ones.
>
> **Summary of Strengths:** Reviewers consistently praised the paper’s clear motivation and conceptual novelty, noting that framing DG as an iterative learn–unlearn process fills an important gap by enabling corrective action after domain-specific bias emerges. The methodological contributions, particularly the principled Inter-Domain Variance (IDV) metric and the coherent USS–IDV pipeline, were highlighted as intuitive and well-justified. Reviewers also emphasized the comprehensive empirical evaluation across fifteen baselines and seven benchmarks, which demonstrates the model-agnostic nature and consistent effectiveness of IU. In addition, the paper’s clarity, organization, and detailed exposition of methods, experiments, and related work were repeatedly commended, along with the strong ablations and analyses that compellingly validate each design choice.
>
> **Summary of Reviewer Concerns & Our Resolutions**
>
> - **Weakness 1: Conceptual Clarity, Intuition, and Robustness of IDV; Clarification of Figure 1**
>   - **Reviewers:** #y5GA-w1, #y5GA-w3, #etHq-w1, #etHq-w2, #etHq-w3, #ZijE-w2, #ZijE-w4, #CNti-w1, #CNti-2
>   - **Resolution:** We clarified that IDV is a domain-aware dispersion metric: it measures the cross-domain inconsistency in activation variance, which directly reflects whether a channel encodes domain-specific behavior. Unlike AV, which pools all samples and mixes within-domain noise with between-domain shifts (and becomes biased under domain imbalance), IDV computes per-domain variances first and then measures their variability across domains, making it naturally robust to domain-size imbalance. The method does not rely on semantic notions such as “texture” or “background”; **Fig. 1** is purely illustrative. Channels are selected solely via statistical IDV + MAD thresholding. We also emphasized empirical robustness: **Fig. 5** shows a clean long-tailed IDV distribution even with a few/imbalanced domains, and Table 4 shows consistent superiority of IDV over AV. We also updated the revision to include the reviewer-requested **DomainDrop experiment (Appendix E, Table 3)**, **DomainDrop t-SNE visualization (Appendix H, Fig. 10)**, and an additional **domain-imbalance illustration (Appendix K, Fig. 12)**.
>
> - **Weakness 2: Distinction from Prior Global Channel-Suppression Methods and Advantages of Post-Epoch IU**
>   - **Reviewers:** #y5GA-w2, #etHq-w4, #CNti-3
>   - **Resolution:** We clarified that prior methods apply **global, mild regularization** to all samples without identifying which ones actually introduce domain-specific bias. In contrast, IU performs **targeted post-epoch unlearning** by jointly localizing both harmful samples (USS) and domain-specific channels (IDV), and applying strong gradient ascent only to their intersection. This difference in granularity explains why globalizing DSGA (ERM_DSCS) erases both harmful and beneficial signals and collapses performance, whereas selective IU consistently improves OOD generalization. IU is fully model-agnostic and orthogonal to existing DG approaches, it modifies neither the loss nor the architecture, and can be plugged on top of them. As requested, we have added the **IU + DomainDrop** results to the revision **(Appendix E, Table 3)**, demonstrating that IU complements, rather than conflicts with, existing channel-modulation methods.

---

> ### Author Response · Authors · 2025-12-02
> **Summary of Rebuttal**
>
> - **Weakness 3: Computational Overhead from Per-Sample Influence Estimation**
>   - **Reviewers:** #oxVH-w1, #oxVH-w4, #ZijE-w1, #YFJ2-w1
>   - **Resolution:** We clarified that IU does **not** conceptually require execution every epoch. Running IU every $e$ epochs (e.g., $e \in [1,5]$) is a natural alternative that reduces the cost by roughly $e\times$ without modifying the algorithm. We also conducted these frequency ablations in the revision **(Appendix J, Table 12)**. We also added the missing ERM per-epoch timing to facilitate direct comparison **(Appendix I, Table 11)**. Additionally, our LiSSA/HVP-based influence estimation is highly parallelizable and can be subsampled, enabling efficient scaling on multi-GPU or newer hardware. These combined strategies substantially mitigate IU's computational overhead while retaining its generalization benefits.
>
> - **Weakness 4: Role of EMA Relative to IU’s Core Contribution**
>   - **Reviewers:** #oxVH-w3
>   - **Resolution:** We clarified that the core contribution is **IU itself** (USS + IDV + DSGA). EMA is a **lightweight, empirically driven refinement** motivated by analyzing the temporal dynamics of unlearning scores. It improves score stability and separability but does not alter the underlying algorithm. We always report **IU and IUE side by side**, with IU serving as the primary method and IUE as an optional stabilized variant.
>
> - **Weakness 5: Conceptual Clarity and Theory–Method Alignment**
>   - **Reviewers:** #etHq-w5, #YFJ2-w3, #CNti-w4, #CNti-w5, #ZijE-w3
>   - **Resolution:** We clarified that the complexity score measures a sample’s influence on parameter change (model complexity), while the generalization score measures its influence on validation loss—a standard first-order influence-based way to capture a sample’s joint effect on parameters and generalization. Theorem 1 motivates IU’s two selective components: IDV-based Domain-Specific Channel Selection (DSCS) approximates the domain-specific parameter subset $\theta_{\text{spc}}$, and Unlearning Set Selection (USS) identifies the harmful-sample subset — conditions not satisfied by generic feature-suppression methods, as confirmed by our USS-only and DSCS-only ablations. Regarding stability, IU updates are localized (i.e., a small subset of channels and samples) and re-selected via MAD at each epoch, while EMA smooths noisy score trajectories. Together, these mechanisms prevent drift from accumulating across epochs. We also clarified the terminology distinction between **task-level unlearning of domain-specific reliance** and privacy-driven data deletion in the **related work section**.
>
> - **Weakness 6: Robustness of IDV and Necessity of Joint Sample+Channel Selectivity**
>   - **Reviewers:** #oxVH-w2, #YFJ2-w2
>   - **Resolution:** We clarified that IDV’s role is **localization**, identifying where domain-specific information resides, not determining how much unlearning should be applied. The strength and direction of unlearning come from USS. ERM_DSCS and ERM_USS are intentionally strong controls: the former applies DSGA globally across all samples, and the latter across all channels within the unlearning set. Both erase useful invariant structure and thus collapse performance, consistent with prior observations that indiscriminate suppression harms representation capacity. These results demonstrate that IDV is stable, but **effective unlearning requires joint selectivity over both samples (USS) and channels (IDV)**, which is exactly the mechanism implemented in IU.

---

### Meta-Review · Area_Chair_ZS43 · 2026-01-05

**Summary:**

The submission received mixed scores (2, 4, 4, 6, 6, 6), but the consensus among the positive reviewers (oxVH, ZijE, CNti) recognizes the novelty of the "post-epoch unlearning" for DG. The primary friction points were the computational cost and the relationship with existing methods like DomainDrop. Reviewer etHq (Score 2) raised concerns about the alignment between the illustrative motivation (texture/background) and the specific mechanics of the method.

**Reviewer Concerns:**

The authors provided a critical new experiment (Table 3) showing that their method consistently improves upon DomainDrop. This demonstrates that IU is complementary and synergistic, directly addressing the concerns of Reviewer y5Ga regarding novelty and redundancy. The computational cost concerns were also mitigated by new ablations. Reviewer etHq remained critical of the illustrative nature of Figure 1.  However, I find that the intuition behind the method stands despite the figure being an abstraction, and the strong empirical consistency across baselines outweighs this critique.

**Reviewer Scores:**

I thought Reviewer y5Ga (Score 4) would likely have flipped to an accept, as the primary condition (comparison with DomainDrop) was met with positive results showing synergy. The three positive reviewers (Scores 6) would likely have solidified their support given the new evidence of complementarity. Reviewer etHq might have maintained their score,  but there is already consensus on acceptance established elsewhere.

---

### Decision · Program_Chairs · 2026-01-26

Accept (Poster)